# Fetal-to-fetal kidney transplantation in utero
Keita Morimoto [1], Shuichiro Yamanaka [1,2] ✉, Kenji Matsui [1], Yoshitaka Kinoshita [1,3], Yuka Inage [1,4], Shutaro Yamamoto[5], Nagisa Koda[1], Naoto Matsumoto[1], Yatsumu Saito[1], Tsuyoshi Takamura[1], Toshinari Fujimoto [1], Shohei Fukunaga[1], Susumu Tajiri[1], Kei Matsumoto[1], Katsusuke Ozawa[6], Seiji Wada[6], Eiji Kobayashi[7] & Takashi Yokoo [1] ✉

Potter sequence consists of various symptoms associated with renal dysplasia. For bilateral renal agenesis, there is no hope of survival. As a novel therapeutic approach for Potter sequence, we develop a unique approach of "transplantation of fetal kidneys from a different species during the fetal stage." In this study, we first validate the approach using allogeneic transplantation. Fetal kidneys with bladders from green fluorescent protein-expressing rats (embryonic day 14.0–16.5) are subcutaneously transplanted into allogeneic rat fetuses in utero (embryonic day 18.0–18.5). After birth, the transplanted fetal kidneys are confirmed to have urine production capability. Furthermore, long-term (up to 150 days) urine production is sustained. Next, we perform xenotransplantation. The transplantation of mouse fetal kidneys into rat fetuses in utero leads the maturation of renal tissue structures. We demonstrate organ transplantation into in utero fetuses using fetal kidneys as donor organs for fetal therapy.

Potter sequence, which is characterized by bilateral renal dysplasia, oligo-hydramnios, and consequent pulmonary hypoplasia, is one of the most serious perinatal diseases[1–4]. Bilateral renal dysplasia resulting in Potter sequence occurs in 1 in 4000 individuals[5,6]. In cases with bilateral renal agenesis, either fetal death occurs or the affected infants die within 12 h after birth due to pulmonary hypoplasia in all cases[6]. In other words, without intervention, the neonatal mortality rate is 100%[7]. Even if pulmonary hypoplasia is prevented through continuous amniotic fluid infusion therapy and immediate postnatal death is avoided, dialysis therapy is essential from birth. However, because of their prematurity and associated abdominal complications, these infants often fail to reach a stage at which dialysis can be safely initiated and sustained, leading to an exceedingly high mortality rate from this condition[8]. Therefore, there is hopeful anticipation that interventions serving as a bridge to achieve a state where dialysis can be safely performed will markedly improve life expectancy. Currently, there is no established effective treatment for such severe fetal renal failure, and dialysis therapy is performed as a limited treatment option after birth.

We have previously reported that rat fetal kidneys can grow, differentiate, and produce urine in the body (retroperitoneal space) of adult

rats[9–13]. Similar phenomena have been confirmed in interspecies combinations, such as rat–mouse and pig–mouse pairs[14,15]. We have developed a method of transplantation of the fetal kidney, ureter, and bladder unit as an approach to avoid hydronephrosis for a limited period[9]. This urological unit is called fetal kidneys with bladders (metanephroi with bladders: MNBs).

Fetal kidneys at this stage of development lack the necessary glomerular structure with vascular invasion for use as donor organs in our studies[16,17]. After transplantation, recipient-derived blood vessels invade the transplanted fetal kidneys and form glomeruli. The vascular endothelium is at the forefront of rejection in organ transplantation, and the advantage of constructing the vascular endothelium of the transplanted organ using self-vessels is that these vessels are less likely to be the target of rejection[18,19]. For fetal treatment, the current kidney transplantation with vascular anastomosis does not fit the organ size and seems impossible as a fetal surgical technique. In contrast, fetal kidneys are appropriately sized and can be transplanted without vascular anastomosis. Therefore, we considered transplanting xenogeneic fetal kidneys with bladders into intrauterine fetuses as a bridge therapy until dialysis can be stably performed.

[1]Division of Nephrology and Hypertension, Department of Internal Medicine, The Jikei University School of Medicine, Tokyo, 105-8461, Japan. [2]Kidney Applied Regenerative Medicine, Project Research Units, The Jikei University School of Medicine, Tokyo, 105-8461, Japan. [3]Department of Urology, Graduate School of Medicine, The University of Tokyo, Tokyo, 113-8654, Japan. [4]Department of Pediatrics, The Jikei University School of Medicine, Tokyo, 105-8461, Japan. [5]Department of Urology, The Jikei University School of Medicine, Tokyo, 105-8461, Japan. [6]Center for Maternal-Fetal, Neonatal and Reproductive Medicine, National Center for Child Health and Development, Tokyo, 157-8535, Japan. [7]Department of Kidney Regenerative Medicine, The Jikei University School of Medicine, Tokyo, 105-8461, Japan. ✉e-mail: shu.yamanaka@jikei.ac.jp; tyokoo@jikei.ac.jp

However, it is unclear whether MNBs can be transplanted, engrafted, and mature in the fetus that is targeted for treatment, with no such reported cases being available in the literature. In this study, we transplanted rat MNBs into the subcutaneous tissue of rat fetuses and confirmed their engraftment, maturation, and urine production. We also investigated the immunological advantages of using fetuses as recipients at the developmental stage of the immune system. In addition, the model of xenogeneic transplantation was validated by transplanting mouse fetal kidneys into rat fetuses, which confirmed the maturation of the transplanted kidneys and yielded less tissue damage caused by rejection compared to the transplantation of mouse fetal kidneys into adult rats.

We aimed to develop a novel treatment for fetuses with bilateral renal agenesis, a condition currently without treatment, and have conceived a concept of fetal therapy—the transplantation of xenogeneic fetal kidneys with bladders during the fetal stage. This study demonstrates the effectiveness of organ transplantation during the fetal stage and is expected to contribute to the development of groundbreaking treatments for congenital kidney diseases, such as the Potter sequence. Although there have been reports of the transplantation of smaller cells into the amniotic fluid[20], peritoneal cavity[21,22], or retroperitoneal space[23] of intrauterine fetuses, to the best of our knowledge, there have been no reports of the transplantation of larger tissue bodies, such as organs, into intrauterine fetuses accompanied by the endowment of function, including in rodent models. In this study, we successfully transplanted fetal organs during the fetal stage and conducted an analysis of the transplanted tissue.

## Results

### Development of methods for fetal-to-fetal transplantation

We developed a method for transplanting MNBs from green fluorescent protein (GFP)-Sprague–Dawley (SD) rats into the subcutaneous space of rat fetuses in utero. An overview of the experiment is provided in Fig. 1a. One MNB extracted from a GFP-SD rat (embryonic day 14.0–16.5 (E14.0–16.5)) (Fig. 1b) was loaded onto the tip of a 15–16 G needle (Saito Medical Instruments Inc., Tokyo, Japan). Pregnant SD rats (E18.0–18.5) were anesthetized via isoflurane inhalation. Subsequently, a midline incision was made, and the uterus was gently extracted from the abdominal cavity. The position of the fetuses was confirmed through the semitransparent uterine wall (Fig. 1c). The uterine wall was punctured with the 15–16 G needle loaded with GFP rat MNBs on the dorsal side of the fetus, and the needle was further inserted into the fetal subcutaneous space while being monitored under a stereomicroscope (Fig. 1d, e; Supplementary Movie 1 online). The needle was advanced approximately 5 mm under the fetal skin, and 0.1–0.5 mL of Hank's balanced salt solution (HBSS, TERUMO, Tokyo, Japan) was ejected to transplant the GFP rat MNBs loaded onto the needle tip into the fetal subcutaneous space. Immediately after transplantation, the needle was withdrawn. The presence of GFP-positive tissue was confirmed by fluorescence stereomicroscopy immediately after transplantation to ensure proper execution (Fig. 1f).

To minimize the duration of surgery, 2–4 fetuses per mother were used as recipients (approximately 40 min of total anesthesia time was required for transplantation into the 4 fetuses). Approximately 5 mL of HBSS warmed to approximately 30 °C was administered intraperitoneally to prevent adhesions. The anatomical positioning of the uterus within the abdominal cavity

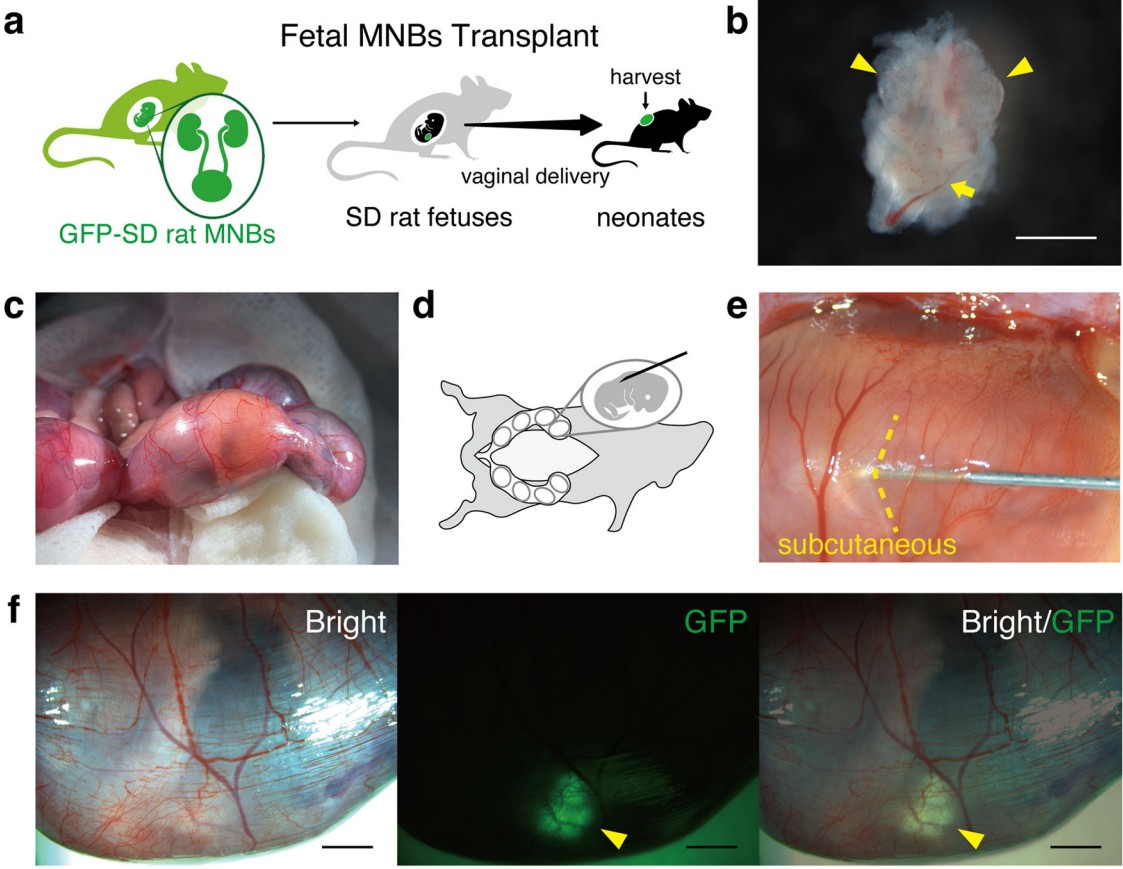

**Fig. 1 | Development of methods for fetal-to-fetal transplantation. a** Summary of the experiment. Transplantation of GFP-SD rat MNBs (E14.0–16.5) into SD rat fetuses and harvest of MNBs after birth. **b** Representation of the comprehensive profile of GFP-SD rat MNBs (E14.5). MNBs are a structure consisting of two fetal kidneys (metanephroi, arrowhead) and a bladder (arrow). **c** The position of the fetuses was identified through the translucent uterine wall. **d** Depiction of the exposure of the uterus and the puncturing of the fetus. **e** The needle was inserted directly into the fetal subcutaneous space. **f** The presence of GFP-positive tissue (arrowhead) was confirmed by fluorescence stereomicroscopy immediately after transplantation. Scale bars, 1 mm in (**b**); 2 mm in (**f**). GFP, green fluorescent protein.

## Table 1 | Transplant success rate

| Pregnant rat number | Number of live pups/ all fetuses | Survival rate [%] | Number of transplanted fetuses | Number of GFP-positive neonates | Success rate (%) |
|---|---|---|---|---|---|
| 1 | 12/14 | 86 | 2 | 1 | 50 |
| 2 | 9/11 | 82 | 4[a] | 4 | 100 |
| 3 | 11/14 | 79 | 2 | 2 | 100 |
| 4 | 8/14 | 57 | 1[a] | 1 | 100 |
| Average | – | 76 | – | – | 88 |

[a]The difficulty of the intrauterine transplantation procedure varies depending on the orientation and size of the fetus and the amount of amniotic fluid. In pregnant rat number 4, the transplantation technique was challenging, requiring extended time to transplant into a single fetus. To avoid prolonged anesthesia time, we completed the procedure after transplanting into just one fetus. In contrast, in pregnant rat number 2, the transplantation procedure was relatively straightforward, allowing us to successfully transplant into four fetuses.

was confirmed, and the incision was closed. Four days after transplantation, on E22, natural delivery occurred, and live offspring were successfully obtained. By puncturing the subcutaneous space of the fetuses in utero, the invasiveness of the procedure was reduced, and the fetuses survived the invasiveness of the thick 15-16 G needle.

We performed transplantation on fetuses in four pregnant rats and calculated the transplantation success rate. The survival rate through natural delivery after transplantation surgery was 75% (40 of 53 fetuses from four pregnant rats survived; Table 1). On the day of natural delivery, the backs of all offspring were photographed using a fluorescence stereomicroscope to confirm the presence of GFP-positive tissue. Among the transplanted fetuses, GFP-positive tissue was confirmed in offspring with an average transplant success rate of 89% (Eight out of nine fetuses from four pregnant rats were successfully transplanted, Table 1).

### Evaluation of transplanted GFP-SD rat MNB maturation in neonates

After natural delivery on embryonic day 22, 4 days after the transplantation of the GFP-SD rat MNBs, the neonates were nursed by their biological mothers and grew. There was no significant difference in body weight gain between the transplanted ($n = 3$) and nontransplanted ($n = 3$) neonates, suggesting the absence of growth impairment associated with the transplantation (Fig. 2a). GFP-positive tissue was observed on the body surface of the neonates using a fluorescence stereomicroscope up to 28 days posttransplantation (24 days after birth), and a gradual increase in the size of the transplanted MNBs was noted (Fig. 2b, c). On day 28 posttransplantation, the dorsal epidermis was incised, and the transplanted MNBs were examined, confirming the invasion of recipient subcutaneous blood vessels into the transplanted MNBs (Fig. 2d; Supplementary Fig. 1 online for the structure of the fetal kidney before transplantation). Percutaneous ultrasonography revealed the presence of a urinary cyst (Fig. 2e), suggesting that the transplanted MNBs were producing urine. Histologically, glomeruli containing nephrin-positive cells, lotus tetragonolobus lectin (LTL)-positive proximal tubules, and E-cadherin (ECAD)-positive distal tubules were observed, confirming the maturation of the MNBs (Fig. 2f, g). In contrast, when GFP-SD rat MNBs were transplanted into adult SD rats and the tissue was retrieved 14 days posttransplantation, rejection was observed and consequently, glomerular and tubular structures could not be identified, with marked infiltration of CD3-positive T cells ($n = 1$, Fig. 2h). Compared with transplantation into adults where glomerular and tubular structures could not be identified at all, MNB transplantation into fetuses demonstrated attenuated rejection, suggesting that MNB transplantation into fetuses has immunological advantages regarding GFP antigen and allogeneic grafts. Moreover, GFP-negative/CD31-positive vascular endothelial cells were detected, suggesting that recipient-derived vessels invaded the transplanted structures and formed glomeruli, producing urine (Fig. 2i, j,

images showing transplantation of E14.5 fetal kidneys). Conversely, GFP-positive CD31-positive blood vessels were also observed, indicating the presence of donor-derived vessels (Fig. 2k showing the transplantation of E14.5 fetal kidneys).

To evaluate the chimeric vessels, we assessed the proportions of recipient- and donor-derived vessels. GFP-positive/CD31-positive, i.e., donor-derived, vascular endothelial cells were observed in all glomeruli, whereas GFP-negative/CD31-positive, i.e., recipient-derived, vascular endothelial cells were detected in 45% of the glomeruli (Table 2). These findings demonstrated the successful creation of a mature exogenous kidney by transplanting allogeneic MNBs into rat fetuses in utero.

### Development of the urine excretion method for GFP-SD rat MNBs transplanted into the subcutaneous space

GFP-SD rat MNBs were transplanted into SD rat fetuses. Four days after transplantation, on embryonic day 22, natural delivery occurred. At 22 days posttransplantation (18 days after birth), a clearly raised structure, presumably the transplanted MNBs, was observed from the body surface (Fig. 3a). Percutaneous ultrasonography revealed a distinct accumulation of fluid with a long diameter of approximately 1 cm, indicating a sufficient urine production capacity for safe percutaneous aspiration (Fig. 3b). Subsequently, a single puncture was performed on the back of the animals using a 23–29 G needle, and urine was aspirated (Fig. 3c; Supplementary Movie 2 online). Thereafter, single punctures were repeated 1–2 times per week, and continuous and stable urine excretion to the outside of the body was successfully maintained up to 150 days posttransplantation (146 days after birth) (Fig. 3d). On average, it was shown that approximately 1 mL of urine was produced per day. In addition, the measurement of creatinine clearance (CCr) showed that it ranged from approximately 40 to 80 μL/min, confirming that the produced fluid was urine with solute-removal capacity ($n = 1$, Fig. 3d; Supplementary Table 2). After confirming long-term urine excretion, tissue retrieval was performed at 150 days posttransplantation (146 days after birth) (Fig. 3e, f). Compared with the tissue image obtained at 28 days posttransplantation (Fig. 2f), a more extensive mature tubular structure was observed (Fig. 3g).

To evaluate the efficacy of aspiration puncture in preventing hydronephrosis, we compared tissues with and without aspiration puncture. For the group without aspiration puncture, GFP-SD rat MNBs were transplanted into SD rat fetuses and retrieved 78 days posttransplantation ($n = 1$). The earlier timepoint of 78 days was selected as the approximate middle timepoint of the 150-day observation period, aiming to avoid severe hydronephrosis that would cause thinning of the renal parenchyma and lead to unclear histologic findings. Masson's trichrome staining revealed extensive fibrotic areas in the nonaspirated tissue (Fig. 3h). In contrast, the tissue retrieved with regular aspiration 150 days posttransplantation did not exhibit fibrotic areas ($n = 1$; Fig. 3h). These findings demonstrated that repeated aspiration punctures performed 1–2 times/week were sufficient to prevent hydronephrosis. Therefore, it was confirmed that administering repeated aspiration punctures for urine excretion prevented hydronephrosis and allowed longer-term maturation of the MNBs. To investigate whether the frequency of 1–2 punctures per week was appropriate for optimal urine production, the correlation between the urine storage time between punctures (in this study, the minimum urine storage time between punctures was 24 h) and the amount of urine produced per 24 h was examined. It was found that shorter urine storage times tended to result in a higher urine output per 24 h ($n = 1$. Fig. 3i).

To ensure reproducibility, similar MNB transplantation experiments into fetuses were performed 17 times. However, in allogeneic transplantation, only 23.5% (4/17) of the individuals developed a urinary cyst of sufficient size (a major diameter of at least 1 cm on ultrasonography), as described above. In all experiments, tissue retrieval was performed more than 3 weeks after transplantation, and it was inferred that rejection occurred to varying degrees in individuals with poor urine production. This suggests that fetal transplantation exhibits varying degrees of immunological advantage acquisition.

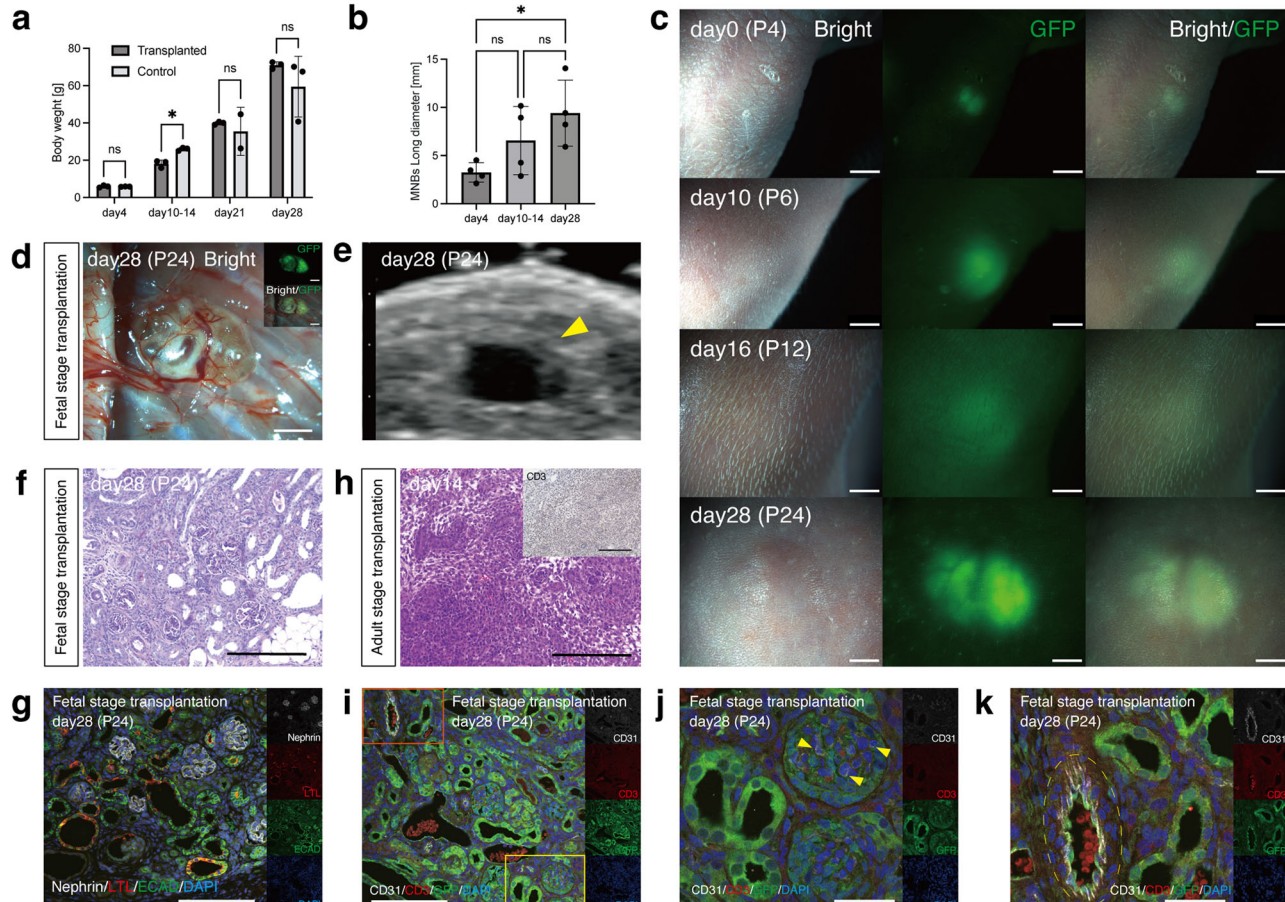

**Fig. 2 | Evaluation of transplanted GFP-SD rat MNB maturation in neonates.**
**a** After natural delivery on embryonic day 22, there was no significant difference in body weight gain between the transplanted ($n = 3$) and nontransplanted ($n = 3$) neonates. **b, c** GFP-positive tissue was observed on the body surface of the neonates and a gradual increase in the size of the transplanted MNBs was noted ($n = 4$). All data are presented as the mean ± standard error of the mean. The data were analyzed using the two-tailed unpaired $t$-test. $*P < 0.05$. **d** On day 28 posttransplantation, recipient subcutaneous blood vessels had invaded the transplanted MNBs. **e** Percutaneous ultrasonography revealed the presence of a urinary cyst (arrowhead), suggesting that the transplanted MNBs were producing urine. **f** On day 28 post-transplantation, mature glomerular and tubular structures were confirmed by periodic acid-Schiff staining. **g** Histologically, glomeruli containing nephrin-positive cells, LTL-positive proximal tubules, and ECAD-positive distal tubules were observed, confirming the maturation of the MNBs. **h** Hematoxylin and eosin staining and CD3 immunohistochemical staining. When GFP-SD rat MNBs were transplanted into adult SD rats and the tissue was retrieved at 14 days post-transplantation, rejection was observed to the extent that glomerular and tubular structures could not be identified, with marked infiltration of CD3-positive T cells ($n = 1$). **i** The yellow frame represents Fig. 2j, whereas the orange frame represents Fig. 2k (images showing transplantation of E14.5 fetal kidneys). **j** The vessels within the glomerulus (CD31-positive) were GFP-negative (arrowhead), indicating recipient-derived vessels. **k** The vessels in the interstitial area (CD31-positive) were GFP-positive (yellow frame), indicating donor-derived vessels. Scale bars, 2 mm in (**c**) and (**d**); 200 μm in (**f**), (**g**), (**h**), and (**i**); 50 μm in (**j**) and (**k**). GFP, green fluorescent protein; LTL, Lotus tetragonolobus lectin; ECAD, E-cadherin.

**Table 2 | Vascular chimerism analysis**

| Experiment number | Number of glomeruli containing GFP-negative/CD31-positive vascular endothelial cells (recipient-derived) | Number of glomeruli containing GFP-positive/CD31-positive vascular endothelial cells (donor-derived) |
|---|---|---|
| Rat (E14.0–E14.5) fetal kidneys were transplanted into rat fetuses (E18.0–E18.5) | | |
| 1 | 6/20 (30%) | 20/20 (100%) |
| 2 | 12/20 (60%) | 20/20 (100%) |
| Total | 18/40 (45%) | 40/40 (100%) |
| Mouse (E13.0) fetal kidneys were transplanted into rat fetuses (E18.0–E18.5) | | |
| 1 | 20/20 (100%) | 10/20 (50%) |
| 2 | 20/20 (100%) | 2/20 (10%) |
| Total | 40/40 (100%) | 12/40 (30%) |

## Attenuation of rejection reaction in secondary fetal kidney transplantation

Using fetuses with developing immune systems as recipients may have immunological advantages. To investigate whether immunological tolerance could be induced as one of the underlying mechanisms, the following experiment was conducted. One fetal kidney was transplanted during the fetal stage, whereas another fetal kidney from the same donor was cryopreserved and subsequently transplanted into the same recipient during the adult stage. Fetal kidneys were individually extracted from GFP-SD rats (E14.0–14.5). One fetal kidney was transplanted into an SD rat fetus in utero (E18.0–18.5). These SD strains have been maintained in an out bred condition. The other fetal kidney from the same individual was then cryopreserved using a previously reported protocol[24,25]. Four days after transplantation, on embryonic day 22, natural delivery occurred, and the pups were nursed by their mothers. One month later, the pups were weaned, and at 9 weeks after birth, when the immune system was fully mature, the

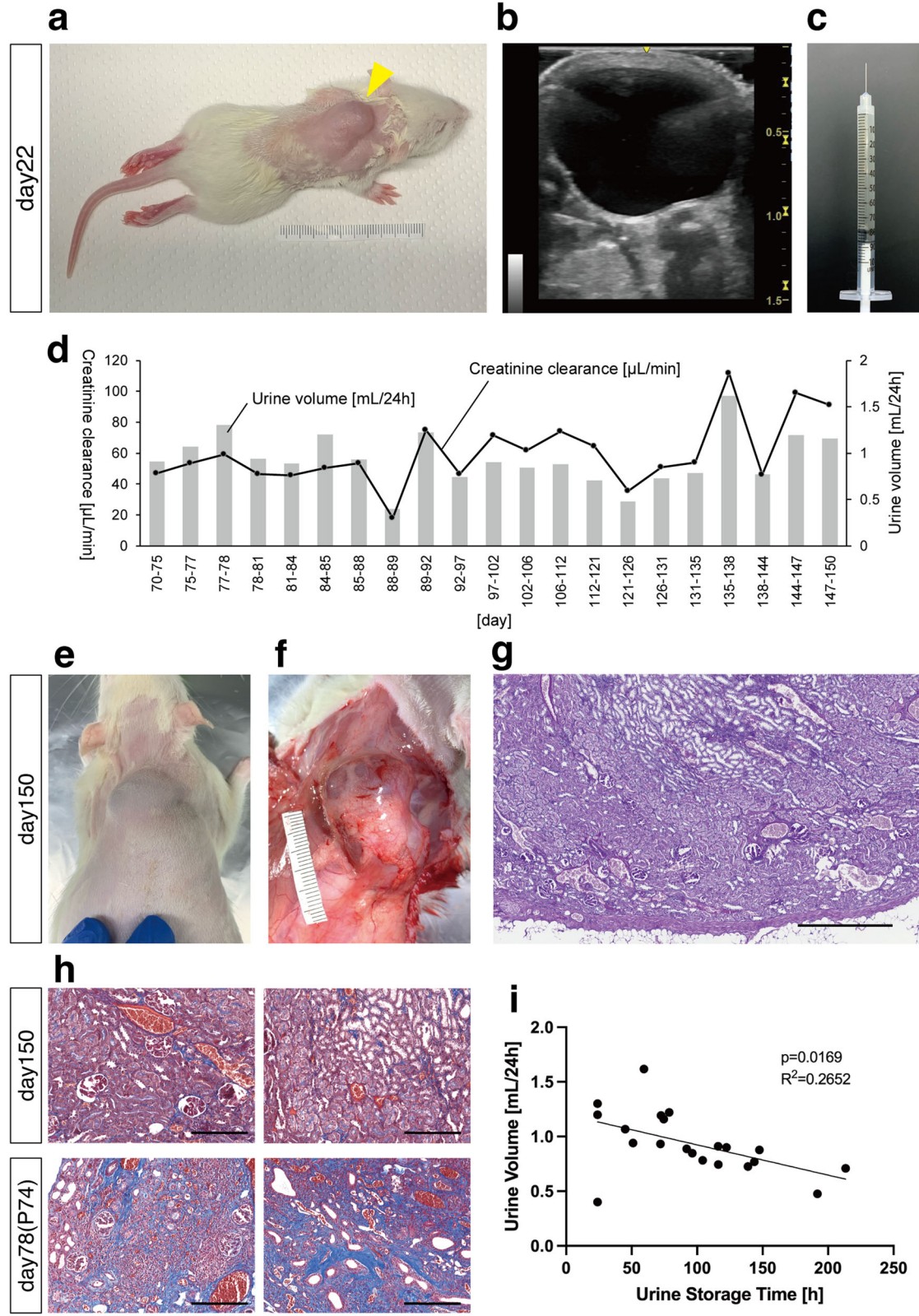

cryopreserved fetal kidney was thawed and retransplanted into the retroperitoneal cavity of each individual, respectively. Tissue retrieval was performed 2 weeks after transplantation (i.e., 11 weeks after birth) (Fig. 4a). In the tissue, the GFP-positive areas were sufficiently preserved, and recipient-derived blood vessels invading the fetal kidneys were observed (Fig. 4b, c). Histologically, minimal lymphocyte infiltration was observed, and the

glomerular and tubular structures were confirmed ($n = 1$, Fig. 4d–f). In contrast, when GFP-SD rat MNBs were transplanted into adult SD rats (at 8 weeks of age) and tissue was retrieved 2 weeks after transplantation (Fig. 2h), glomerular and tubular structures could not be identified due to severe rejection. Therefore, the rejection reaction was attenuated in recipients who received a second fetal kidney transplantation after initial

**Fig. 3 | Development of the urine excretion method for GFP-SD rat MNBs transplanted into the subcutaneous space. a** At 22 days posttransplantation, a clearly raised structure, presumably the transplanted MNBs (arrowhead), was observed on the body surface. **b** Percutaneous ultrasonography revealed distinct fluid accumulation. **c** A single puncture was performed on the back using a 29 G needle, and urine was aspirated. **d** Bar graphs illustrating the volume of urine aspirated via puncture from the transplanted MNBs at a frequency of 1–2 times per week, from day 70 to day 150 posttransplantation. Line graphs depicting the calculated creatinine clearance ($n = 1$). **e, f** Tissue retrieval was performed at 150 days posttransplantation. **g** Using periodic acid-Schiff staining, mature renal tissue structures including glomeruli and more developed tubules were confirmed. **h** Masson's trichrome staining revealed a lack of fibrotic areas at 150 days posttransplantation ($n = 1$) compared with tissue retrieved at 78 days posttransplantation ($n = 1$) without any aspiration punctures. **i** Replotting of the data presented in Fig. 3d as a scatter plot. The data were analyzed using linear regression. Because the minimum urine storage time between punctures was 24 h herein, no data points were available for <24 h ($n = 1$). A $P$-value < 0.05 was considered statistically significant. Scale bars, 500 μm in (**g**); 200 μm in (**h**).

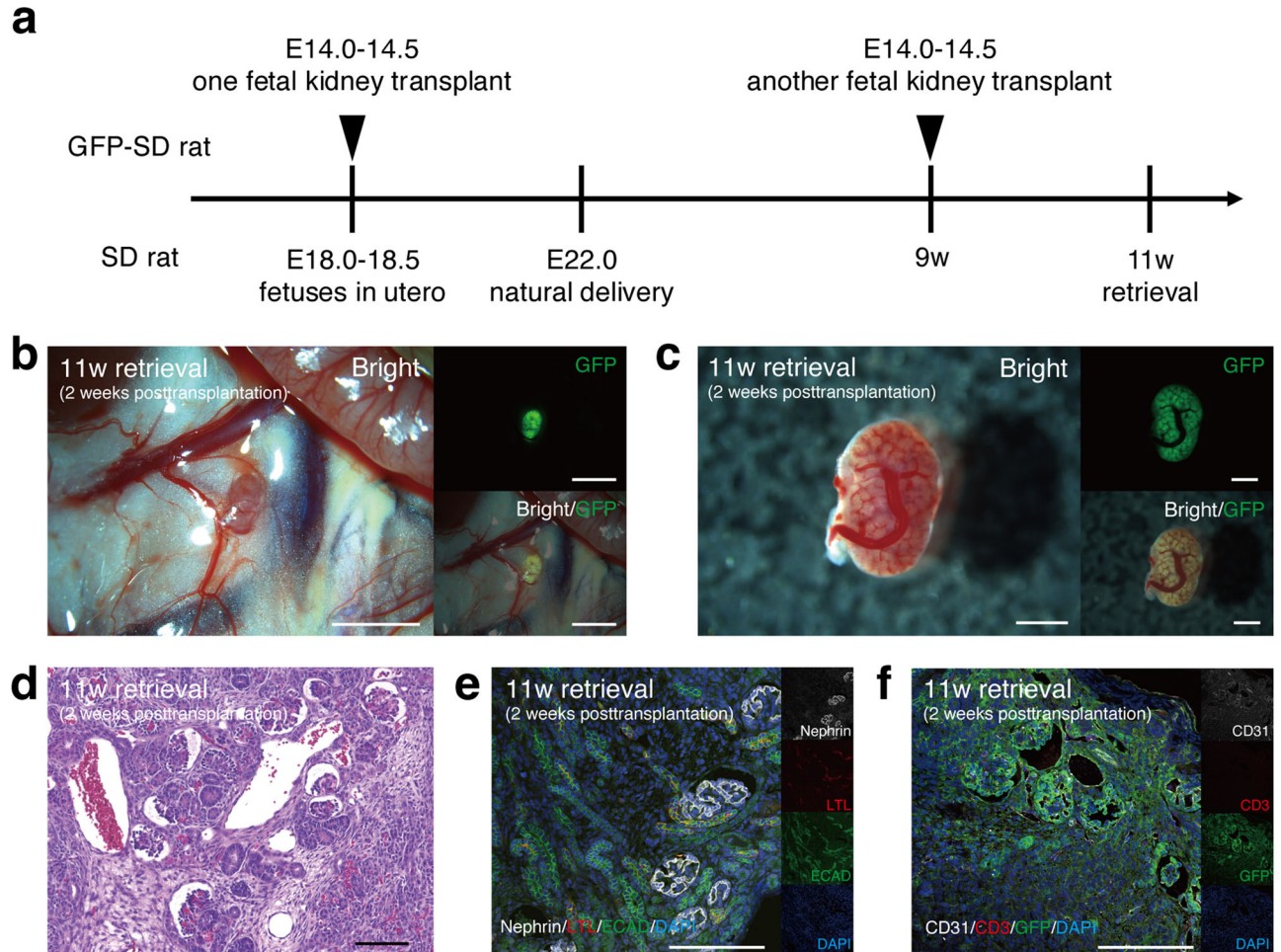

**Fig. 4 | Attenuation of rejection reaction in secondary fetal kidney transplantation. a** Schematic diagram of the experiment. When transplanting one fetal kidney (E14.0–14.5), the other fetal kidney was kept frozen. Nine weeks after transplantation, the frozen kidney was thawed, retransplanted, and tissues were retrieved 11 weeks after the first transplantation. **b, c** In the tissue that was retrieved 2 weeks after transplantation (at 9 weeks of age), the GFP-positive areas were sufficiently preserved, and recipient-derived blood vessels were observed invading the fetal kidneys. **d, e, f** Periodic acid-Schiff staining and immunofluorescence demonstrated that lymphocyte infiltration was minimal, and more glomerular and tubular structures were confirmed compared with the 3-week retrieval ($n = 1$). Scale bars, 2 mm in (**b**); 1 mm in (**c**); 100 μm in (**d**); 200 μm in (**e**) and (**f**).

transplantation during the fetal stage (Fig. 4b–f), compared to those that received their first transplantation during the adult stage (Fig. 2h). However, the rejection reaction in the secondary transplantation, albeit attenuated, implied that complete immunological tolerance was not induced. Although further research is necessary to elucidate the immunological advantages of transplantation into fetuses, these results may help in understanding the underlying mechanisms.

### Verification of xenotransplantation (mice to rats)
A similar experiment was conducted using xenotransplantation of GFP-B6 (C57BL/6) mouse fetal kidneys (E13.0–13.5) into SD rat fetuses (E18.0–18.5). Four days after transplantation, on embryonic day 22, natural delivery occurred, and the pups were nursed by their mothers and grew. Tissue retrieval was performed after transplantation. Even without immunosuppressant administration, in the case of the transplantation into SD rat fetuses, the GFP-positive areas persisted 10 days after transplantation, and numerous CD3-positive cells infiltrated the tubular interstitial region; however, glomerular structures were observed ($n = 1$, Fig. 5a, b). However, 18 days after transplantation, the GFP-positive areas were decreased, severe inflammatory cell infiltration was observed, and no glomerular structures were detected ($n = 1$, Fig. 5c). In contrast, when GFP-B6 mouse fetal kidneys were transplanted into adult SD rats at 8 weeks of age and tissue was retrieved 3 days after transplantation, severe rejection was observed before maturation (Fig. 5d), suggesting that rejection is somewhat reduced even in

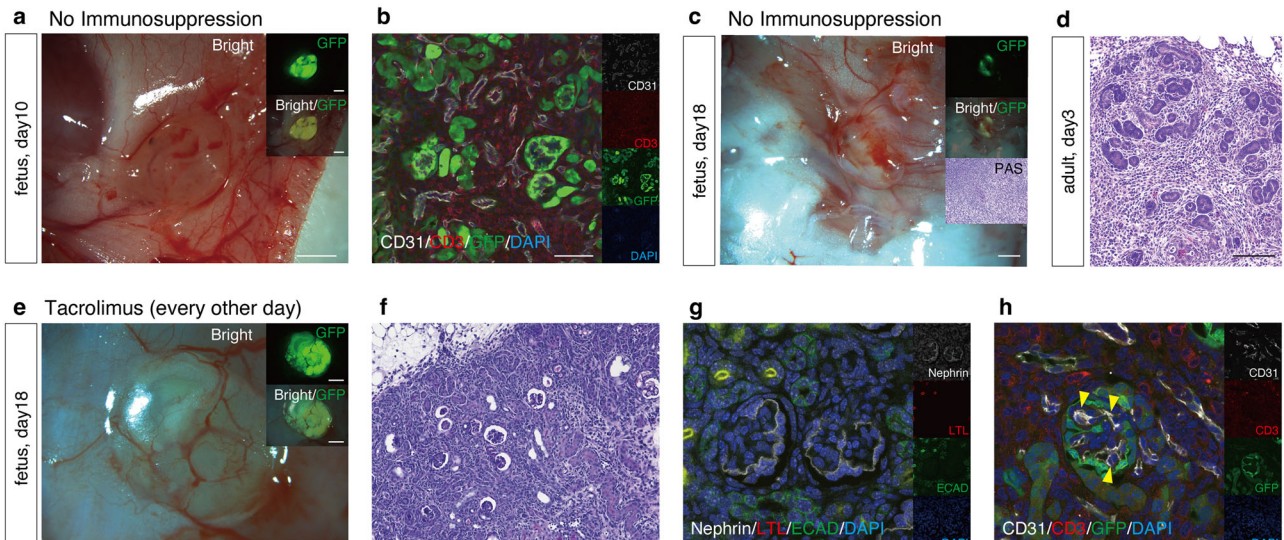

**Fig. 5 | Verification of xenotransplantation (mice to rats). a, b** No immunosuppression. Even without the administration of an immunosuppressant, in the transplantation into SD rat fetuses, the GFP-positive areas persisted 10 days after transplantation, and numerous CD3-positive cells infiltrated the tubular interstitial region; however, glomerular structures were observed ($n = 1$). **c** No immunosuppression. Eighteen days after transplantation, the GFP-positive areas decreased, severe inflammatory cell infiltration was observed, and no glomerular structures were detected (periodic acid-Schiff staining). **d** No immunosuppression. Hematoxylin and eosin staining showed that, when a GFP-B6 mouse fetal kidney was transplanted into adult SD rats at 8 weeks of age and tissue was retrieved 3 days after transplantation, severe rejection was observed before maturation ($n = 1$). **e** After tacrolimus administration, the GFP-positive areas were sufficiently preserved, even at 18 days after transplantation. **f** Periodic acid-Schiff staining revealed the presence of glomerular and partial tubular structures ($n = 2$). **g** Immunostaining revealed ECAD-positive distal tubules. **h** The CD31-positive vessels detected inside the GFP-positive glomerular structures were GFP-negative (arrow), indicating that the glomeruli were completely composed of recipient (mouse)-derived vessels. Marked infiltration of CD3-positive cells was observed in the tubular interstitial region. GFP, green fluorescent protein; PAS, Periodic acid-Schiff; ECAD, E-cadherin. Scale bars, 1 mm in (**a**), (**c**), and (**e**); 100 μm in (**b**), (**d**), and (**f**); 50 μm in (**g**) and (**h**).

xenotransplantation into fetuses. To reduce rejection, tacrolimus (Prograf® injection 2 mg/0.2 mL, Astellas Pharma Inc.) administration was performed. Furthermore, to transfer tacrolimus to the fetuses via the placenta, tacrolimus was subcutaneously administered to the pregnant rats at 0.2 mg/kg/day (every other day, day 0 and 2) after transplantation, and, after birth, the neonates themselves were subcutaneously administered tacrolimus at 0.1 mg/kg/day (every other day, starting from postnatal day 0, then day 2, day 4, and so on). In a preliminary evaluation, the placental transfer rate of tacrolimus from the pregnant rat to the fetuses was approximately 40% (Supplementary Table 1). In the presence of tacrolimus administration, the GFP-positive areas were sufficiently preserved even at 18 days after transplantation ($n = 2$, Fig. 5e); histologically, glomerular and tubular structures were observed (Fig. 5f). Immunostaining revealed the presence of ECAD-positive distal tubules (Fig. 5g). The CD31-positive vessels observed inside the GFP-positive glomerular structures were GFP-negative, indicating that the glomeruli were completely composed of the recipient (mouse)-derived vessels (Fig. 5h). GFP-negative/CD31-positive recipient-derived vascular endothelial cells were observed in all glomeruli, whereas GFP-positive/CD31-positive donor-derived vascular endothelial cells were detected in 30% of the glomeruli (Table 2 and Supplementary Fig. 2). Conversely, marked infiltration of CD3-positive cells was observed in the tubular interstitial region, confirming that tacrolimus monotherapy was insufficient to suppress rejection in the context of xenotransplantation (Fig. 5h).

## Discussion

In this study, we successfully transplanted urological units, i.e., MNBs consisting of a kidney, ureter, and bladder, into rat fetuses in utero using a needle via a transuterine approach. We obtained live offspring through natural delivery and created a mature exogenous kidney.

We focused on subcutaneous transplantation as the transplantation site. In previous studies using adult recipients in rodents, the retroperitoneal space was generally selected as the transplantation site. However, transplantation into the retroperitoneal space requires a laparotomy, which renders it unrealistic to perform transuterine transplantation. Interestingly,

we discovered that subcutaneous transplantation[26], which was considered disadvantageous in adult transplantation, was suitable as a site for fetal organs during fetal transplantation. We have previously attempted to transplant fetal kidneys into the subcutaneous space of adults[26]; however, in all cases, the development was poor, and no significant urine production was obtained. Subcutaneous transplantation had a low incidence of bleeding complications, and the fetal survival rate was high, at 76%. Expanding the transplantation bed as the fetus grows may also alleviate physical constraints. In addition to the physical and spatial advantages, we hypothesized that there might be specific advantages of the fetal subcutaneous space, as there was a remarkable difference in the development of the transplanted MNBs after transplantation, with urine production, compared to adult recipient subcutaneous transplantation.

It has been reported that the composition of the extracellular matrix (ECM) differs between the subcutaneous space of adults and fetuses[27,28]. The ECM promotes the development of angiogenesis in kidney organoids that mimic embryonic organs[29]. In addition, some components of the ECM have been reported to play a role in directing the migration of endothelial cells, contributing to angiogenesis[30]. Thus, an analysis focusing on the unique ECM composition of the fetal subcutaneous space is necessary to understand the difference between adult and fetal subcutaneous transplantation.

The large amount of urine produced for such a long period by a single MNB recorded here has not been observed in retroperitoneal transplantation in adults. We also consider that the ease of urine excretion management through subcutaneous transplantation was an important advantage, as regular urine excretion is crucial for preventing hydronephrosis in the setting of fetal kidney transplantation. Once urine is produced by the fetal kidneys, if left untreated, the transplanted MNBs will develop hydronephrosis because of the lack of a urine excretion pathway, and eventually, renal function will be lost. Therefore, the formation of a urine excretion pathway becomes necessary at 3–4 weeks after transplantation, when urine production is observed. In previous studies, MNBs were transplanted into the retroperitoneal space of adults, making it difficult to excrete urine through percutaneous aspiration from the body surface. Therefore, a

surgical method was employed to connect the urinary cyst to the host's ureter by laparotomy[9]. In our model, we took advantage of the fact that the transplanted MNBs were located subcutaneously and in close proximity to the body surface. We successfully performed safer and continuous percutaneous aspiration for 150 days, thus avoiding hydronephrosis and achieving urine excretion to the outside of the body.

After urine aspiration, it reaccumulates in the bladder over a certain period, but the optimal frequency of aspiration punctures for better urine production was unknown. Therefore, we investigated the appropriate frequency for administering aspiration punctures and found that shorter urine storage times between punctures resulted in larger urine production. This suggests that more continuous urine excretion, rather than intermittent excretion, may result in a higher urine output per unit of time. Considering safety, we performed aspiration punctures 1–2 times per week in this study; however, in the future, we aim to establish a method for continuous drainage, such as placing a catheter.

Moreover, when considering long-term engraftment, it is important to factors in the immunological advantages, especially in this rodent system. In the case of the allogeneic transplantation of MNBs into fetuses (SD rat–SD rat), the rejection was reduced to such an extent that urine production persisted for 150 days without the use of immunosuppressants. This may be attributed to two advantages of fetal kidney transplantation into fetuses in utero. One is that the donor organ used here was a fetal kidney, as previously reported[31,32], and the other is that the recipient was a fetus, a characteristic specific to this study. Regarding the former, it has been reported that fetal kidneys may have a lower immunogenicity than adult-type kidneys[31–36], and the lack of mature antigen-presenting cells in fetal kidneys is a proposed mechanism[33]. Regarding the latter, the intrauterine environment reportedly supports the suppression of immune rejection. For example, the programmed death-1/programmed death ligand 1 pathway inhibits T-cell activation in the placenta[37]. Additionally, HLA-G expression by placental cells suppresses the proliferation and cytotoxicity of T and natural killer cells and promotes the expansion of regulatory T cells[38]. Furthermore, placental cells secrete chemokines and cytokines that shift the Th1/Th2 balance toward Th2 cells and suppress proinflammatory Th17 responses[39]. Collectively, these mechanisms regulate maternal–fetal immune tolerance. This immunologically privileged intrauterine environment may suppress rejection responses against the transplanted MNBs.

Furthermore, transplanted fetal kidneys comprise recipient-derived vessels[40–44], it is at least conceivable that rejection is reduced compared with organ transplantation using adult organs as donors. Even with the transplantation of rat fetal kidneys at E14.5, at a stage preceding the formation of distinct vascular structures, the vasculature was not entirely derived from recipient vessels. Vascular progenitor cells are present within the fetal kidney before the emergence of identifiable vascular structures[40]. These findings suggest that intrinsic vascular progenitor cells present in E14.5 rat fetal kidneys differentiate posttransplantation, thereby developing the chimeric vasculature.

Notably, immunological advantage of this approach was that the recipient was a fetus, a specific characteristic of our technique. Transplantation during the fetal stage, when the immune system is still developing, may have immunological advantages.

In xenogeneic transplantation into fetuses, the addition of a small amount of immunosuppressant resulted in longer-term engraftment (18 days). Previous studies have reported that cell transplantation can engraft in fetuses in utero in xenogeneic settings[45–47]. Although xeno-transplantation faces a higher hurdle in terms of rejection compared with allogeneic transplantation, immunological advantages afforded by the abovementioned mechanisms were speculated.

One of the limitations of this study was that a kidney failure model was not used as the transplantation target. Because all recipient fetuses were normal (wild-type) fetuses, the actual therapeutic effect of fetal kidney transplantation could not be confirmed. The presence of the recipient native kidneys might benefit the transplanted fetal kidneys. Because the recipient kidneys maintained the fluid and electrolyte balance, the transplanted fetal

kidneys were potentially protected from adverse stress, such as hyperfiltration, enabling long-term sustained urine production. Conversely, hyperfiltration developing in the transplanted fetal kidneys would increase GFR and urine output for a short duration. In the future, we plan to perform transplantation verification using a congenital kidney disease model (Six2-expressing nephron progenitor cell depletion model[48]) in rats. Furthermore, the CCr of a single MNB remained low, at 40–80 μL/min, compared with the CCr of 4000 μL/min calculated from the 24 h urine collection of the neonates themselves at the same time point. To achieve therapeutic effects in the future, it is necessary to explore methods to enhance the function of a single MNB or to transplant multiple MNBs. Regarding the technical considerations for clinical application, we have already performed preliminary experiments with the transplantation of porcine fetal kidneys in early gestation (approximately 30 days of gestation) into porcine fetuses in late gestation (approximately 90 days of gestation) to simulate human-sized procedures (unpublished data). We plan to proceed with nonhuman primate studies to evaluate xenograft rejection responses.

The ultimate goal of this study was to develop a renal replacement therapy for children with severe congenital kidney diseases, particularly bilateral renal agenesis, for which there are few treatment options. In this study, we used MNBs as transplantation organs, established a transplantation method into fetuses, and demonstrated for the first time worldwide the success of transplantation into fetuses and the acquisition of functional organs capable of urine production in allogeneic settings. The superiority of fetal tissue as a donor organ has been previously pointed out; however, the present study suggests that the fetal elements on the recipient side may also have immunological and scaffold environment advantages. In this study, we successfully constructed an organ with urine production ability for as long as 150 days, which is considered a sufficient period as a bridge to dialysis in neonates, considering the lifespan of rodents. For application in humans, it is necessary to demonstrate this therapeutic effect using larger animal models, increase the number of transplanted MNBs, and improve the quality of the transplanted MNBs. This study is an important step toward the development of a new treatment approach for medical intervention in children with severe kidney diseases for which no effective treatment is available currently.

## Methods
### Research animals
The animal experiments performed here followed the Guidelines for the Proper Conduct of Animal Experiments of the Science Council of Japan (2006) and were approved by the Institutional Animal Care and Use Committee of the Jikei University School of Medicine (protocol numbers: 2023-021). All efforts were made to minimize animal suffering. Pregnant female SD rats, SD-Tg (CAG-enhanced green fluorescent protein [EGFP]) rats (GFP-SD rats), C57BL/6-Tg (CAG-EGFP) mice (GFP-B6 mice), and adult male SD rats were purchased from Sankyo Labo Service Corporation (Tokyo, Japan).

### Isolation of fetal kidneys or MNBs
Pregnant rats and mice were anesthetized via the inhalation of isoflurane (2817774; Pfizer, New York, USA). The fetal kidneys or MNBs (E14.0–16.5 in rats and E13.0–13.5 in mice) were retrieved, and the pregnant rats and mice were then killed immediately via an infusion of pentobarbital (120 mg/kg). All fetuses were killed by decapitation. Fetal kidneys or MNBs were dissected under a surgical microscope (M205FA; Leica Microsystems, Wetzlar, Germany).

### Methods for fetal-to-fetal transplantation
For details of the transplantation method, please refer to the Results section and Fig. 1.

### Evaluation of transplanted MNB maturation in neonates
After natural birth at E22, the neonates were nursed by their biological mothers in the same cage. At postnatal day 0, the GFP-positive tissue was

visualized on the body surface of the neonates using fluorescence microscopy (M205FA; Leica Microsystems, Wetzlar, Germany). Neonates lacking the GFP-positive tissue were euthanized on the same day by decapitation. Body weight measurements were regularly conducted throughout the observation period. In addition, the GFP-positive tissue was visualized using fluorescence microscopy, and the longitudinal diameter was measured over time using Image J.

### Evaluation of vascular chimerism to distinguish between donor- and recipient-derived vessels

To evaluate chimeric vessels, we assessed the proportions of GFP-negative/CD31-positive (recipient-derived) and GFP-positive/CD31-positive (donor-derived) vessels. In paradigms where E14.0–E14.5 rat and E13.0 mouse fetal kidneys were transplanted into E18.0–E18.5 rat fetuses, we conducted two independent experiments ($n = 2$). Two sections randomly selected from each experiment were used to evaluated the presence of GFP-negative/CD31-positive and GFP-positive/CD31-positive vascular endothelial cells in 10 glomeruli/section.

### Development of the urine excretion method for MNBs transplanted into the subcutaneous space

After natural birth at E22, the transplanted fetal kidneys or MNBs were observed using transcutaneous ultrasonography (LOGIQ e Premium; GE HealthCare Technologies Inc, Chicago, USA). If a urinary cyst with a longitudinal diameter of 5 mm or more was detected via ultrasonography, it was deemed safe for percutaneous aspiration. Neonates were anesthetized by isoflurane inhalation, and a single puncture was made using a 23–29 G needle (TERUMO, Tokyo, Japan) on the dorsal side to aspirate the urine. Single punctures were repeated at a frequency of 1–2 times per week, and the urine volume was measured.

### Cryopreservation and thawing methods for fetal kidneys

The cryopreservation and thawing methods used for fetal kidneys have been reported previously in ref. 24. Initially, the fetal kidneys were equilibrated in a base medium (MEMα supplemented with 20% fetal bovine serum [FBS; SH30070.03, HyClone Laboratories, Inc., Logan, UT, USA] and 1% antibiotic–antimycotic solution [15,240,062; Thermo Fisher Scientific, Waltham, MA, USA]) containing 7.5% ethylene glycol (EG; 055-00996; Wako, Osaka, Japan) and 7.5% dimethyl sulfoxide (DMSO; 317275-100 ML; Millipore, Burlington, MA, USA) on ice for 15 min; followed by soaking in a base medium containing 15% EG and 15% DMSO on ice for an additional 15 min. Subsequently, the fetal kidneys were placed onto Cryotops (81111; Kitazato Corporation, Tokyo, Japan) and immediately submerged into liquid nitrogen for storage until transplantation. Fetal kidneys that had been cryopreserved for several weeks were thawed just prior to transplantation. The Cryotops containing fetal kidneys were swiftly transferred from liquid nitrogen to a base medium containing 1 M sucrose at 42 °C for 1 min, then to a base medium containing 0.5 M sucrose at room temperature for 3 min, and finally washed twice in a base medium at room temperature for 5 min each time.

### In the secondary fetal kidney transplantation performed during the adult stage

**Adult** SD rats were anesthetized via isoflurane inhalation, and a laparotomy was performed through an abdominal midline incision. A pocket was created in the retroperitoneal space within the region bounded by the aorta, left ureter, and left renal artery using microtweezers (11253-25; Dumont, Montignez, Switzerland) under a surgical microscope. A mouse single fetal kidney or rat MNB was transplanted into the pocket, which was subsequently closed using 10-0 nylon sutures (Muranaka Medical Instruments Co. Ltd., Osaka, Japan). The procedure was completed by closing the abdominal incision.

### Biochemical measurements in the blood and urine

For blood tests, SD rats were anesthetized via isoflurane inhalation, and their tail veins were punctured using a 25 G needle (TERUMO) to collect blood samples using hematocrit capillary tubes (2-454-21, AS ONE Corporation). The capillaries were sealed with wax (2-454-22, AS ONE Corporation) and centrifuged at $12,000 \times g$ for 10 min to separate the serum from blood cells. The serum creatinine levels were quantified using a DRI-CHEM instrument (FUJIFILM). Urine samples were collected from MNBs and the rats' 24 h urine collection. For the 24 h urine collection, SD rats were placed in rat metabolic cages (KN-650-MC, Sugiyama-gen Co., Ltd.) overnight. The urine samples were submitted to SRL, Inc. for the measurement of urine creatinine levels. Creatinine clearance was calculated using the following formula:

Creatinine clearance [μL/min] = (urine creatinine levels [mg/dL] × urine volume [μL /min]/serum creatinine levels [mg/dL]).

Blood tacrolimus levels were assayed by electrochemiluminescence immunoassay (ECLIA) using whole blood samples submitted to SRL, Inc.

### Histology and immunofluorescence

The retrieved fetal kidneys or MNBs were fixed in 4% paraformaldehyde (161-20141, Wako) at 4 °C overnight and subsequently embedded in paraffin. Sections with a thickness of 4 μm were obtained using a rotary microtome (HM355S, PHC, Tokyo, Japan). Standard procedures were used for staining with hematoxylin and eosin, Masson's trichrome, and periodic acid-Schiff.

For immunostaining, the slides were deparaffinized, washed thrice in phosphate-buffered saline (PBS), and subjected to antigen retrieval by incubating with citrate buffer (#K 035; 10× citrate buffer, pH 6.0; DBS, Pleasanton, CA, USA) at 121 °C for 10 min. Following three additional washes in PBS, the slides were blocked at room temperature for 30 min using 5.0% skimmed milk. After three washes in PBS, the sections were incubated overnight at 4 °C with primary antibodies (Supplementary Table 3). Following three washes in PBS, the sections were incubated with secondary antibodies conjugated with Alexa Fluor 488, 546, or 647, as well as DAPI for 1 h, at room temperature. After three additional washes in PBS, the sections were mounted with a glycerol-based liquid mounting medium. Each sample was examined under a fluorescence microscope (LSM880; Carl Zeiss, Oberkochen, Germany).

### Immunohistochemistry

Paraffin-embedded sections were deparaffinized and treated with 3% hydrogen peroxide for 20 min, followed by three washes with PBS. For antigen retrieval, the sections were incubated with citrate buffer at 121 °C for 10 min. Next, the sections were washed three times with PBS and blocked with 5.0% skimmed milk at room temperature for 30 min. After three washes with PBS, the sections were incubated with indicated primary antibodies (Supplementary Table 3) at 4 °C overnight, washed three times with PBS, and incubated with appropriate goat antimouse or antirabbit secondary antibodies (Histofine Simple Stain MAX-PO (MULTI) 424151, NICHIREI Biosciences, Japan) at room temperature for 30 min. After three washes with PBS, the staining was visualized by incubating sections with the DAB chromogen (SK-4100, Vector Laboratories, USA) for 1 min. After three washes with distilled water, the sections were counterstained with hematoxylin for 3 s and washed three times with distilled water. Finally, the sections were dehydrated, cleared, and mounted using PathoMount (164-28492, FUJIFILM).

### Statistical analyses

All data are presented as the mean ± standard error of the mean. The data were analyzed using the two-tailed unpaired t-test and linear regression. A P-value of <0.05 was considered statistically significant. Experimental data were analyzed using GraphPad Prism version 8.0 (GraphPad Software, Boston, MA, USA) and Microsoft Excel (Microsoft, Redmond, WA, USA). The details of the analyses are explained in each figure legend.

**Article**

**Reporting summary**

Further information on research design is available in the Nature Portfolio Reporting Summary linked to this article.

## Data availability

The main data supporting the results in this study are available within the paper and its Supplementary Information. All other data are available from the corresponding author upon reasonable requests. Source Data for figures are provided with this paper.

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

## Acknowledgements

We thank Ms. T. Tamatsukuri, Ms. T. Hayakawa, Ms. S. Kawagoe, and Ms. H. Gotoh for their experimental and technical assistance. This work was supported by the Japan Agency for Medical Research and Development (grant no. 22bm0704049h0003, 24bm1223003h0003 and 23bm1123036h0001), the Japan Society for the Promotion of Science (JSPS-KAKENHI; grant no. 21K08288 and 24K11439), and JST FOREST Program (grant no. JPMJFR2011).

## Author contributions

K. Morimoto wrote the main manuscript text. K. Morimoto and S. Yamanaka designed the study. K. Morimoto performed the experiments. K. Morimoto, S. Yamanaka, S. Yamamoto, N.K., Y.K., Y.I., K. Matsui, N.M., Y.S., T.T., T.F., S.F., S.T., K. Matsumoto, K.O., S.W., E.K., T.Y. interpreted the data and revised the manuscript. S. Yamanaka and T.Y. supervised the study. All authors reviewed the manuscript and granted permission to publish the study. T.Y. granted final permission to the manuscript.

## Competing interests

Author Eiji Kobayashi is the director of Kobayashi Regenerative Research Institute, LLC. Eiji Kobayashi has received a research fund from Sumitomo Pharma Co., Ltd., Osaka, Japan as a result of the Collaborative Research Agreement between The Jikei University School of Medicine and Sumitomo Pharma Co., Ltd. Author Eiji Kobayashi have received technical guidance fee from Fuji Micra Inc., Shizuoka, Japan. The authors declare that this research was conducted in the absence of any commercial or financial relationships that could be construed as a potential conflict of interest. The funders had no role in the design of the study; in the collection, analyses, or interpretation of data; in the writing of the manuscript, or in the decision to publish the results.
