## [Transparent Peer Review file · Communications Biology]

Fetal-to-Fetal Kidney Transplantation in Utero

Corresponding Author: Professor Takashi Yokoo

Version 0:

Reviewer comments:

Reviewer #1

(Remarks to the Author)

This manuscript aims to translate human fetal kidney transplantation towards renal replacement therapy in bilateral renal agenesis. The authors use rodent fetal kidneys as donors and for hosts allogeneic and xenogeneic fetal and adult rodent recipients. They do not perform animal experiments in the setting they actually wish to cure, e.g. potter sequence/renal agenesis.

The technical aspects are very interesting and worthwhile.

Comments:

1. The idea of this investigation is that at the end human fetal kidneys will be utilized as donors in utero. Unfortunately, none of the earlier work on using grafts of human fetal kidney is mentioned in this study. Experiments with human fetal kidneys grafted into immunodeficient rodent hosts that were also reconstituted with human immune system showed the immunological advantage of using fetal kidneys over adult kidneys (Dekel B, Transplantation 1997; 64:1541-1550; Dekel B, Transplantation 1997; 64:1550-1558). Furthermore, molecular mechanisms of immune evasion by human fetal kidneys were delineated (Dekel B, Transplantation 2000; 69:1470-8). These authors could also show that developing human fetal kidney grafts follow similar gene expression observed during nephrogenesis (Dekel B, J Am Soc Nephrol 2002;13:977-90) and that there is a window of opportunity of human and pig metanephric grafts, affording the best post-transplant growth, optimal host-donor vascularization and the least immunogenicity (Dekel B, Nat Med 2003;9:53-60). These earlier studies are at the basis of this translational study and serve its purpose in terms of advantages of the source tissue and should be cited.

2. The authors use E14-E16.5 kidney transplants as donors. E14 represent the earliest timepoint. This is crucial since glomeruli at this point are yet to be vascularized. It would be therefore critical to understand if there is a specific day affording enhanced growth, less immunogenicity and importantly host versus donor vascularization. Can glomeruli which are yet vascularized at E14-E15 be vascularized only by host capillaries growing inside? it is reasonable that vascularized glomeruli at E16.5 will show only donor type vascularization and will therefore be more immunogenic. All of these questions are important for rejection aspects and need to be addressed for specific embryonic days.

3. Pls elaborate more on the clinical scenario. If there is intention to transplant potter in utero, one needs to find an aborted fetu. That is Of course complicated in many ways. Is there rationale to divert to pig fetal kidneys for in utero transplants as envisioned and studied in Nat Med 2003;9:53-60?

Reviewer #2

(Remarks to the Author)

The paper by Dr. Morimoto et al entitled "Fetal Kidney Transplantation Into In Utero Fetuses is an interesting and well written manuscript. The goal of this paper is to begin to evaluate a novel approach to address the rare fetal condition of renal agenesis that is fatal when it develops bilaterally. There is no effective treatment for this condition. The models used were 1. Allogeneic transplantation of green fluorescent protein (GFP) labeled rat fetal kidney tissue consisting of metanephroi with ureter and bladder (MNBs) into a fetus of a pregnant SD rat without GFP labeling and 2. Xenogeneic transplantation of murine GFP labeled fetal MNBs into non-labeled xenogeneic rat fetuses in utero. Both of these models used critical timing targeting pre-glomerular development of the MNBs where renal development would theoretically continue after transplantation. Similarly, the recipients were at the cusp of immune system development. Both the time factors of the donors and recipients employed could theoretically impact the experimental outcomes. Some discussion of the potential impact

would be of interest.

A flaw inherent in using these models is that the recipient fetuses retained their native renal tissue. The authors did not address the potential impact of the native recipient kidneys during the posttransplant monitoring period of up to 5 months in the allogeneic model. In many previous transplantation studies the native kidneys are nephrectomized in order to be able to ascertain the renal function of the transplanted tissue. While nephrectomy of the native fetal tissue in the recipient rats many have complicated the surgical aspects of the model used in this study, the authors should at least discuss the impact of having native renal function in the recipients during the posttransplant follow up period.

The Discussion section of the manuscript is largely a repeat of the data in the Results section, with a short discussion of a potential role for the ECM as a basis for their differing results between adult and fetal recipients. The authors state “the ultimate goal of this study was to develop a renal replacement therapy for children with severe congenital kidney diseases, particularly bilateral renal agenesis...” The paper would be significantly enhanced if there was a discussion of how they envision such a therapy, along with the technical hurdles that will need to be overcome. A particularly relevant issue would be the potential need for chronic immunosuppression in children at such a young age in the case of transplanting xenogeneic tissue. It appears that the authors' vision is for the technology to be merely a bridge to dialysis and if the patient is very lucky a kidney allograft.

Specific issues:

Page 5 – It is stated that the in utero environment appeared to provide protection from rejection. It is well known that the uterine environment is largely immune privileged. Citing this literature would strengthen the interpretation of the results. Many decades ago the issue of whether recipient or donor cells recolonized the vasculature of the an allograft was researched. The consensus was that the kidney tissue remained comprised of donor cells. How this previous data, in light of the observed GFP negative, CD31+ cells along the vasculature of the transplanted tissue, strengthens your interpretation of the MNBs protection being provided by the in utero environment would be of interest.

The authors stated that the SK rats were outbred. However, no genotyping was provided. If the recipients and donors were in fact syngeneic then the inherent histocompatibility could have prevented rejection and impacted the MNBs viability outcomes.

Page 5 – There is no mention of the recipient serum creatinine values, just the creatinine value aspirated from the renal cysts. It is assumed that this omission is because the native kidneys provided normal renal function in the neonatal rat recipients. Therefore, the functional potential of the MNBs in supporting the rats has not been established.

Page 7 - Cryopreservation is well known to compromise the viability of mammalian tissues. The authors state “the rejection observed in the tissue that was retrieved 2 weeks after the transplantation of GFP-SD rat MNBs into adult SD rats (at 8 weeks of age) in which glomerular and tubular structures could not be identified, the rejection reaction was weak.” Was the viability of the cryopreserved MNBs confirmed prior to transplantation?

Similarly, “when GFP-B6 mouse fetal kidneys were transplanted into adult SD rats at 8 weeks.... Severe rejection was observed before maturation”.... While it is stated in the Discussion that the murine MNBs were placed into the retroperitoneal space, how were the murine MNBs transplanted?

Discussion – “We obtained live offspring through natural delivery and created a mature exogenous kidney.” The degree of maturation of the MNBs is not well defined, in conjunction with the presence of the native kidneys, the claim of producing a mature exogenous kidney is not well supported.

Page 13 – Stated that the GFP+ tissue was visualized using fluorescence microscopy. How was the fluorescence quantified?

Page 14 – Since the MNBs were not vascularized, their viability must have been dependent upon diffusion of the amniotic fluid. Where any controls utilized where comparative testing of the amniotic fluid was conducted?

Page 15 – Immunofluorescence: the protocol does not include where a washing step was utilized after incubations with the primary and secondary antibodies. If the slides were not copiously washed to remove any unbound antibody the results could have been impacted.

Table 1 – In Case 2 why were 4 transplants performed and only 1 in Case 4?

Supplemental Table 1 – An explanation for the two dosing regimens and their relative impact on outcomes could potentially strengthen your interpretation of the results.

Page 25, Figure 2 – What software was utilized to objectively quantify the green fluorescence is warranted

Reviewer #3

(Remarks to the Author)

Comments for Authors

I would like to congratulate the authors on an excellent and innovative manuscript. Their work demonstrates a high level of creativity and technical expertise, addressing a critical and challenging medical need with a novel therapeutic approach. The concept of transplanting fetal kidneys from different species during the fetal period is both groundbreaking and ambitious, offering potential insights into solutions for conditions such as Potter sequence. The experimental design and the demonstration of long-term functionality of transplanted organs reflect a high level of technical skill and commitment to advancing this field.

That said, there are some aspects of the manuscript that could be improved to enhance clarity. I recommend the following:

1. Title Revision: Consider revising the title to highlight the novel concept of fetal-to-fetal transplantation. This will better reflect the unique and groundbreaking aspect of your study.

2. Align the Methods and Results sections: Ensure that the titles and organization of the Methods subsections correspond directly to the subsections in the Results and Figures. This will create a logical flow and help readers connect the experimental design with the findings.

3. Separate Methods from Results: Remove any methodological details currently described under the Results section and integrate them into the appropriate parts of the Methods section. See for example the first subsection of the Results with methodological details already appearing in the Methods.
4. Provide Sample Sizes (n): Clearly indicate the sample size (n) for all experiments and analyses, both in the text and figure legends, to allow readers to assess the robustness of your findings.
5. Clarify Table 1: The term "Case" (e.g., Case 1 to Case 4) in Table 1 requires clarification. Provide a detailed explanation of what each "Case" represents, including any distinctions between them. Include a comprehensive description of the data in Table 1 within the text. Expand Table if necessary. Add table legend.
6. Tacrolimus Details. Clarify the type of drug used (include the manufacturer), add a reference or detailed description of the protocol used for administering tacrolimus. Describe the method or assay used to detect and measure tacrolimus levels.
7. Line 133 – replace “indicating” with “suggesting”
8. Figure 2h: Figure 2h appears to demonstrate cellular infiltration. Consider providing additional data, such as specific staining to identify the phenotype of the infiltrating cells and clarify their role in the observed results.
9. Data on Rejection Rates: In line 148, the statement that "MNB transplantation into fetuses yielded a reduced rate of rejection" lacks supporting data. Please clarify.
10. CD31-Positive Images: Clarify how many experiments were performed and how many slides were stained in each experiment to ensure reproducibility and reliability of the results. Provide additional representative images if possible.
11. Correct typo in Fig.2 legend: 2j should be replaced with 2k
12. Line 197: Verification of the immunological advantages of fetal transplantation. This section describes a novel approach - two kidneys from the same fetal donor were transplanted into the same recipient at two different time points – fetal and adult. This should be better clarified in the title and in the text of this section. Please also better clarify the goal of this experiment.
13. Clarify Data Presentation in Figure 4b-f: indicate which image corresponds to each experimental group, clearly describe the experimental groups, including retrieval times (two-week was compared to three week? – please clarify)
14. Separate and Describe Adult Transplantation Experiment (Lines 630-632): The description of the experiment involving transplantation into adult recipients should be clearly separated and presented under both the Methods and Results sections. This will ensure clarity and proper distinction from the fetal transplantation experiments.
15. Fig 3d: please clarify that the line represents creatinine clearance and the bars represent urine volume. Provide details on how creatinine clearance was calculated. Include data on creatinine and urine volume measurements in the Methods and Results sections.
16. Fig 3h The comparison of day 150 with day 78 without aspiration punctures in Figure 3h represents a standalone experiment. Provide a detailed description of this experiment in both the Methods and Results sections, including the experimental design, sample sizes, and rationale for the comparison.
17. Fig 3i: Ensure consistent terminology throughout the text and figure legend (e.g., "storage time" versus "accumulation between punctures time"). Data for periods shorter than 24 hours do not appear on the plot. Please provide a more comprehensive explanation of the data in Figure 3i, both in the text and figure legend, to ensure readers fully understand the findings.
18. Discussion: Discuss earlier reports, describing the ability of avascular, embryonic kidneys (1,2) and pancreas (3) to attract host vasculature, rendering them less susceptible to humoral rejection (4):
 - (1) Dekel, B. et al. Human and porcine early kidney precursors as a new source for transplantation. *Nat. Med.* <https://doi.org/10.1038/nm812> (2003).
 - (2) Takeda, S. I., Rogers, S. A. & Hammerman, M. R. Differential origin for endothelial and mesangial cells after transplantation of pig fetal renal primordia into rats. *Transpl. Immunol.* <https://doi.org/10.1016/j.trim.2005.10.003> (2006).
 - (3) Hecht, G. et al. Embryonic pig pancreatic tissue for the treatment of diabetes in a nonhuman primate model. *Proc. Natl. Acad. Sci. U.S.A.* <https://doi.org/10.1073/pnas.0812253106> (2009).
 - (4)

Version 1:

Reviewer comments:

Reviewer #1

(Remarks to the Author)

Thanks you for the revised version.

The authors have addressed some of my concerns in the discussion.

They have not added experimental data analyzing age of fetal transplant, host-donor vascularization and transplant rejection status but rather discuss this matter.

They failed to cite important "old" papers that are highly relevant to the translation of fetal kidney transplantation clarifying how fetal kidney transplants sustain a molecular program that supports nephrogenesis in vivo and why fetal kidney transplants avoid immune rejection including

Transplantation. 1997 Dec 15;64(11):1550-8, Transplantation. 2000 Apr 15;69(7):1470-8, J Am Soc Nephrol. 2002 Apr;13(4):977-990. These should be added.

Reviewer #2

(Remarks to the Author)

The authors have extensively revised their manuscript within a quick turnaround time. The issues raised in the critiques from the three previous reviewers have been well addressed; including retitling the manuscript as suggested Reviewer 3. The Methods and Results sections have been better aligned with a much clearer separation of methods from the result sections.

Greater technical detail has also been provided with: 1. clarification of the (n) within the experimental groups, 2. the software used for quantification of fluorescence, 3. more specific details of the experimental protocols used for the studies, 4. use of consistent terminology throughout the text, and 5. the inclusion of additional relevant literature. The edited manuscript reads more clearly along with improved graphs, tables and histological pictures. This is a well written manuscript on a topic that could hold potential for future clinical advancements.

Reviewer #3

(Remarks to the Author)

Dear Editors,

Thank you for the opportunity to review this manuscript. The authors have thoroughly addressed my concerns and have made the necessary revisions to improve clarity and strengthen their work. I believe the manuscript now meets the standards for publication and recommend acceptance.

1/15/2025

Manuscript number: COMMSBIO-24-4847

Manuscript title: Fetal-to-Fetal Kidney Transplantation in Utero

(Manuscript title before revision: Fetal Kidney Transplantation into In Utero Fetuses)

Responses to Reviewers' Comments

We greatly appreciate the reviewers' detailed comments and suggestions. We have revised the manuscript and the figures to address these points. We believe that they have been improved through these revisions. Below are our point-by-point responses to the comments. Responses to the comments are shown in red in the revised manuscript and in this letter.

Reviewer #1:

This manuscript aims to translate human fetal kidney transplantation towards renal replacement therapy in bilateral renal agenesis. The authors use rodent fetal kidneys as donors and for hosts allogeneic and xenogeneic fetal and adult rodent recipients. They do not perform animal experiments in the setting they actually wish to cure, e.g. potter sequence/renal agenesis.

The technical aspects are very interesting and worthwhile.

> Thank you for your valuable comments and suggestions. We will address the validation using the Potter's sequence model as a future research objective.

Comments:

1. The idea of this investigation is that at the end human fetal kidneys will be utilized as donors in utero. Unfortunately, none of the earlier work on using grafts of human fetal kidney is mentioned in this study. Experiments with human fetal kidneys grafted into immunodeficient rodent hosts that were also reconstituted with human immune system showed the immunological advantage of using fetal kidneys over adult kidneys (Dekel B, Transplantation 1997; 64:1541-1550; Dekel B, Transplantation 1997; 64:1550-1558). Furthermore, molecular mechanisms of immune evasion by human fetal kidneys were delineated (Dekel B, Transplantation 2000; 69:1470-8). These authors could also show that developing human fetal kidney grafts follow similar gene expression observed during nephrogenesis (Dekel B, J Am Soc Nephrol 2002;13:977-90) and that there is a window of opportunity of human and pig metanephric grafts, affording the best post-transplant growth, optimal host-donor vascularization and the least immunogenicity (Dekel B, Nat Med 2003;9:53-60). These earlier studies are at the basis of this translational study and serve its purpose in terms of advantages of the source tissue and should be cited.

> Our ultimate therapeutic goal is to transplant "porcine" fetal kidneys into human fetuses, not

human fetal kidneys. As stated in lines 67-68 of our manuscript, "we considered transplanting xenogeneic fetal kidneys with bladders into intrauterine fetuses as a bridge therapy until dialysis can be stably performed." We specifically used the term "xenogeneic fetal kidneys" to reflect this approach. We have added the following text to lines 309-310, citing Dekel B, Nat Med 2003;9:53-60 as reference 33: "the lack of mature antigen-presenting cells in fetal kidneys is a proposed mechanism."

2. The authors use E14-E16.5 kidney transplants as donors. E14 represent the earliest timepoint. This is crucial since glomeruli at this point are yet to be vascularized. It would be therefore critical to understand if there is a specific day affording enhanced growth, less immunogenicity and importantly host versus donor vascularization. Can glomeruli which are yet vascularized at E14-E15 be vascularized only by host capillaries growing inside? it is reasonable that vascularized glomeruli at E16.5 will show only donor type vascularization and will therefore be more immunogenic. All of these questions are important for rejection aspects and need to be addressed for specific embryonic days.

> In Figures 2i, j, and k, we demonstrate the chimeric formation of donor- and recipient-derived vessels using data from transplanted "E14.5" GFP-SD rat fetal kidneys. Our results show that even when fetal kidneys were transplanted at E14.5, before the formation of distinct vascular structures, donor-derived vessels were present post-transplantation. As described in reference 37, although in mice, vascular progenitor cells are present within the fetal kidney as early as E10.5. Therefore, we propose that vascular progenitor cells present in the E14.5 rat fetal kidneys differentiated after transplantation, leading to the formation of chimeric vessels.

We have clarified this in the Results section (lines 144-147) and figure legend (line 650) by adding "images showing transplantation of E14.5 fetal kidneys." Additionally, in the Discussion section (line 321-325), we have added: "Even with the transplantation of rat fetal kidneys at E14.5, at a stage preceding the formation of distinct vascular structures, the vasculature was not entirely derived from recipient vessels. Vascular progenitor cells are present within the fetal kidney before the emergence of identifiable vascular structures³⁷. 7. These findings suggest that intrinsic vascular progenitor cells present in E14.5 rat fetal kidneys differentiate posttransplantation, thereby developing the chimeric vasculature."

3. Pls elaborate more on the clinical scenario. If there is intention to transplant potter in utero, one needs to find an aborted fetus. That is Of course complicated in many ways. Is there rationale to divert to pig fetal kidneys for in utero transplants as envisioned and studied in Nat Med 2003;9:53-60?

> As mentioned in Comment 1, our ultimate goal is to transplant porcine fetal kidneys into human

fetuses. The rationale for using porcine fetal kidneys is well-supported by the paper you referenced.

Reviewer #2:

The paper by Dr. Morimoto et al entitled “Fetal Kidney Transplantation Into In Utero Fetuses is an interesting and well written manuscript. The goal of this paper is to begin to evaluate a novel approach to address the rare fetal condition of renal agenesis that is fatal when if it develops bilaterally. There is no effective treatment for this condition. The models used were 1. Allogeneic transplantation of green fluorescent protein (GFP) labeled rat fetal kidney tissue consisting of metanephroi with ureter and bladder (MNBs) into a fetus of a pregnant SD rat without GFP labeling and 2. Xenogeneic transplantation of murine GFP labeled fetal MNBs into non-labeled xenogeneic rat fetuses in utero. Both of these models used critical timing targeting pre-glomerular development of the MNBs where renal development would theoretically continue after transplantation. Similarly, the recipients were at the cusp of immune system development. Both the time factors of the donors and recipients employed could theoretically impact the experimental outcomes. Some discussion of the potential impact would be of interest.

A flaw inherent in using these models is that the recipient fetuses retained their native renal tissue. The authors did not address the potential impact of the native recipient kidneys during the posttransplant monitoring period of up to 5 months in the allogeneic model. In many previous transplantation studies the native kidneys are nephrectomized in order to be able to ascertain the renal function of the transplanted tissue. While nephrectomy of the native fetal tissue in the recipient rats many have complicated the surgical aspects of the model used in this study, the authors should at least discuss the impact of having native renal function in the recipients during the posttransplant follow up period.

The Discussion section of the manuscript is largely a repeat of the data in the Results section, with a short discussion of a potential role for the ECM as a basis for their differing results between adult and fetal recipients. The authors state “the ultimate goal of this study was to develop a renal replacement therapy for children with severe congenital kidney diseases, particularly bilateral renal agenesis...” The paper would be significantly enhanced if there was a discussion of how they envision such a therapy, along with the technical hurdles that will need to be overcome. A particularly relevant issue would be the potential need for chronic immunosuppression in children at such a young age in the case of transplanting xenogeneic tissue. It appears that the authors’ vision is for the technology to be merely a bridge to dialysis and if the patient is very lucky a kidney allograft.

> Thank you for your valuable comments and suggestions. We will address the validation using

the Potter's sequence model as a future research objective.

As noted in the Limitations section, we did not perform recipient kidney removal because the creatinine clearance from the transplanted fetal kidneys was low, and bilateral nephrectomy of the recipient's own kidneys would likely have resulted in death.

The presence of the recipient native kidneys might benefit the transplanted fetal kidneys. Because the recipient kidneys maintained the fluid and electrolyte balance, the transplanted fetal kidneys were potentially protected from adverse stress, such as hyperfiltration, enabling long-term sustained urine production. Conversely, hyperfiltration developing in the transplanted fetal kidneys would increase GFR and urine output for a short duration. This has been added to line 327-342 of the manuscript.

Regarding the technical considerations for clinical application, we have already performed preliminary experiments with the transplantation of porcine fetal kidneys in early gestation (approximately 30 days of gestation) into porcine fetuses in late gestation (approximately 90 days of gestation) to simulate human-sized procedures (unpublished data). We plan to proceed with nonhuman primate studies to evaluate xenograft rejection responses. This has been added to line 347-352 of the manuscript.

In response to the comment that "It appears that the authors' vision is for the technology to be merely a bridge to dialysis and if the patient is very lucky a kidney allograft," we emphasize that our target population consists of fetuses with bilateral renal agenesis, the most severe form of kidney disease. These fetuses currently cannot survive due to their inability to undergo dialysis given their overall condition. Therefore, even as a bridge to dialysis, this treatment represents a potentially life-saving intervention for fetuses who currently have no chance of survival under existing medical care.

Specific issues:

Page 5 – It is stated that the in utero environment appeared to provide protection from rejection. It is well known that the uterine environment is largely immune privileged. Citing this literature would strengthen the interpretation of the results. Many decades ago the issue of whether recipient or donor cells recolonized the vasculature of the an allograft was researched. The consensus was that the kidney tissue remained comprised of donor cells. How this previous data, in light of the observed GFP negative, CD31+ cells along the vasculature of the transplanted tissue, strengthens your interpretation of the MNBs protection being provided by the in utero environment would be of interest.

> We have added the following text to line 310-318 of the manuscript, citing previous studies to support how the immune-privileged intrauterine environment likely contributed favorably to our results.

"Regarding the latter, the intrauterine environment reportedly supports the suppression of immune rejection. For example, the programmed death-1/programmed death ligand 1 pathway inhibits T-cell activation in the placenta³⁴. Additionally, HLA-G expression by placental cells suppresses the proliferation and cytotoxicity of T and natural killer cells and promotes the expansion of regulatory T cells³⁵. Furthermore, placental cells secrete chemokines and cytokines that shift the Th1/Th2 balance toward Th2 cells and suppress proinflammatory Th17 responses³⁶. Collectively, these mechanisms regulate maternal–fetal immune tolerance. This immunologically privileged intrauterine environment may suppress rejection responses against the transplanted MNBs."

The authors stated that the SK rats were outbred. However, no genotyping was provided. If the recipients and donors were in fact syngeneic then the inherent histocompatibility could have prevented rejection and impacted the MNBs viability outcomes.

> As you correctly point out, we unfortunately did not evaluate genotypes in this experiment. While there may be individuals with coincidentally similar genetic backgrounds, SD rats are generally considered outbred, suggesting that transplantation without immunosuppression would typically result in rejection. Furthermore, in addition to using SD rats, the donor organs expressed GFP protein, which itself can elicit strong rejection responses. This is demonstrated in our control experiment shown in Figure 2h, where GFP-SD rat fetal kidneys were transplanted into adult SD rats.

Page 5 – There is no mention of the recipient serum creatinine values, just the creatinine value aspirated from the renal cysts. It is assumed that this omission is because the native kidneys provided normal renal function in the neonatal rat recipients. Therefore, the functional potential of the MNBs in supporting the rats has not been established.

> In Figure 3d, we present creatinine clearance calculations using the recipient's serum creatinine levels. While the creatinine clearance calculated from the urine volume produced by MNBs, urinary creatinine levels, and recipient's serum creatinine levels is not sufficient to sustain life independently, the non-zero values indicate that the MNBs are functionally active. We have added the creatinine clearance calculation method to line 449-452 of the Methods section. Additionally, we have included the actual measured values of urine volume, urinary creatinine concentration, and recipient serum creatinine concentration as Supplementary Table 2.

Page 7 - Cryopreservation is well known to compromise the viability of mammalian tissues. The authors state "the rejection observed in the tissue that was retrieved 2 weeks after the transplantation of GFP-SD rat MNBs into adult SD rats (at 8 weeks of age) in which glomerular and tubular structures could not be identified, the rejection reaction was weak." Was the viability

of the cryopreserved MNBs confirmed prior to transplantation?

> Although we did not specifically evaluate the survival rate of fetal kidneys before and after freezing in this experiment, our previous studies (references 24 and 25) have demonstrated that the freezing preservation method used for MNBs does not compromise their survival rate. Therefore, we consider it self-evident that the freezing preservation method used for MNBs in this study maintains their viability.

Similarly, “when GFP-B6 mouse fetal kidneys were transplanted into adult SD rats at 8 weeks.... Severe rejection was observed before maturation”.... While it is stated in the Discussion that the murine MNBs were placed into the retroperitoneal space, how were the murine MNBs transplanted?

The method for mouse fetal kidney transplantation is described in the Methods section (line 432-439).

Discussion – “We obtained live offspring through natural delivery and created a mature exogenous kidney.” The degree of maturation of the MNBs is not well defined, in conjunction with the presence of the native kidneys, the claim of producing a mature exogenous kidney is not well supported.

> In Figure 2g, we present immunostaining of transplanted tissue retrieved 28 days after transplanting rat MNBs into rat fetuses. Nephryn, LTL, and ECAD are widely recognized as maturation markers: Nephryn for podocytes in glomeruli, LTL for proximal tubules, and ECAD for distal tubules. The clear positive staining for these maturation markers demonstrates that the transplanted tissue has achieved maturation.

Page 13 – Stated that the GFP+ tissue was visualized using fluorescence microscopy. How was the fluorescence quantified?

> We measured the diameter using ImageJ based on images captured with fluorescence microscopy (M205FA; Leica Microsystems, Wetzlar, Germany). We have added the following text to line 393-395 of the Methods section: "In addition, the GFP-positive tissue was visualized using fluorescence microscopy, and the longitudinal diameter was measured over time using Image J."

Page 14 – Since the MNBs were not vascularized, their viability must have been dependent upon diffusion of the amniotic fluid. Where any controls utilized where comparative testing of the amniotic fluid was conducted?

> Thank you for your comment. I apologize, but I am not entirely clear about your question. Could

you please explain it again?

Page 15 – Immunofluorescence: the protocol does not include where a washing step was utilized after incubations with the primary and secondary antibodies. If the slides were not copiously washed to remove any unbound antibody the results could have been impacted.

> We did perform the washing steps. We apologize for this omission. We have added to line 464-468 of the Methods section that three additional washes in PBS were performed before primary antibody incubation and before and after secondary antibody incubation.

Table 1 – In Case 2 why were 4 transplants performed and only 1 in Case 4?

> "The difficulty of the intrauterine transplantation procedure varies depending on the orientation and size of the fetus and the amount of amniotic fluid. In Case 4 (Pregnant rat number 4), the transplantation technique was challenging, requiring extended time to transplant into a single fetus. To avoid prolonged anesthesia time, we completed the procedure after transplanting into just one fetus. In contrast, in Case 2 (Pregnant rat number 2), the transplantation procedure was relatively straightforward, allowing us to successfully transplant into four fetuses." The above text has been added as a footnote to Table 1 (line 721-727).

Supplemental Table 1 – An explanation for the two dosing regimens and their relative impact on outcomes could potentially strengthen your interpretation of the results.

> The administration regimen has been added as supplementary text to Supplemental Table 1 (line 735-736).

Page 25, Figure 2 – What software was utilized to objectively quantify the green fluorescence is warranted

> We measured the diameter using ImageJ. We have added the following text to line 395 of the Methods section: "using Image J."

Reviewer #3:

Comments for Authors

I would like to congratulate the authors on an excellent and innovative manuscript. Their work demonstrates a high level of creativity and technical expertise, addressing a critical and challenging medical need with a novel therapeutic approach. The concept of transplanting fetal kidneys from different species during the fetal period is both groundbreaking and ambitious, offering potential insights into solutions for conditions such as Potter sequence. The experimental

design and the demonstration of long-term functionality of transplanted organs reflect a high level of technical skill and commitment to advancing this field.

That said, there are some aspects of the manuscript that could be improved to enhance clarity. I recommend the following:

> Thank you for your kind words. Please find our responses to your points below.

1. Title Revision: Consider revising the title to highlight the novel concept of fetal-to-fetal transplantation. This will better reflect the unique and groundbreaking aspect of your study.

> We have changed the title from "Fetal Kidney Transplantation into In Utero Fetuses" to "Fetal-to-Fetal Kidney Transplantation in Utero" to better convey the fetal-to-fetal nature of the transplantation.

2. Align the Methods and Results sections: Ensure that the titles and organization of the Methods subsections correspond directly to the subsections in the Results and Figures. This will create a logical flow and help readers connect the experimental design with the findings.

> We have aligned the titles of the Results and Methods sections to correspond with each other as follows:

Results: Development of methods for fetal-to-fetal transplantation (line 90)

Methods: Methods for fetal-to-fetal transplantation (line 385)

Results: Evaluation of transplanted GFP-SD rat MNB maturation in neonates (line 122)

Methods 1: Evaluation of transplanted MNB maturation in neonates (line 388)

Methods 2: Evaluation of vascular chimerism to distinguish between donor-derived and recipient-derived vessels (line 397)

Results: Development of the urine excretion method for GFP-SD rat MNBs transplanted into the subcutaneous space (line 154-155)

Methods: Development of the urine excretion method for MNBs transplanted into the subcutaneous space (line 405-406)

Results: Attenuation of rejection reaction in secondary fetal kidney transplantation (line 196)

Methods: In the secondary fetal kidney transplantation performed during the adult stage (line 432)

3. Separate Methods from Results: Remove any methodological details currently described under the Results section and integrate them into the appropriate parts of the Methods section. See for example the first subsection of the Results with methodological details already appearing in the Methods.

> Since this is a novel transplantation technique that has never been performed before, we

consider the establishment of the methodology itself to be one of our results. Therefore, we have maintained the detailed description in the Results section, while the Methods section now simply states: " For details of the transplantation method, please refer to the Results section and Fig. 1." (line 386).

4. Provide Sample Sizes (n): Clearly indicate the sample size (n) for all experiments and analyses, both in the text and figure legends, to allow readers to assess the robustness of your findings.

> We have added the n values at the relevant locations, specifically on lines 139, 174, 179, 212, 230, 232, 401, 649, 666, 667, 670, 688, 693, 695, and 703)

5. Clarify Table 1: The term "Case" (e.g., Case 1 to Case 4) in Table 1 requires clarification. Provide a detailed explanation of what each "Case" represents, including any distinctions between them. Include a comprehensive description of the data in Table 1 within the text. Expand Table if necessary. Add table legend.

> We have clarified the term "case" which referred to pregnant rats. We added the following text to line 114-115: "We performed transplantation on fetuses in four pregnant rats and calculated the transplantation success rate. Additionally, in Table 1, we have replaced "Case" with "Pregnant rat number" and listed them as 1, 2, 3, and 4.

6. Tacrolimus Details. Clarify the type of drug used (include the manufacturer), add a reference or detailed description of the protocol used for administering tacrolimus. Describe the method or assay used to detect and measure tacrolimus levels.

> We have added the following details to the manuscript:

In the Results section:

•Product name: "Prograf® injection 2 mg/0.2 mL, Astellas Pharma Inc." (line 236)

•Administration protocol: "every other day, day 0 and 2" (line 238)

•Detailed schedule: "starting from postnatal day 0, then day 2, day 4, and so on" (line 240-241)

In the Methods section:

• Measurement method: "Blood tacrolimus concentrations were assayed by electrochemiluminescence immunoassay (ECLIA) using whole blood samples submitted to SRL, Inc." (line 453-454)

7. Line 133 – replace “indicating” with “suggesting”

> This has been corrected (line 126).

8. Figure 2h: Figure 2h appears to demonstrate cellular infiltration. Consider providing additional data, such as specific staining to identify the phenotype of the infiltrating cells and clarify their role in the observed results.

> We have added CD3 staining to Figure 2h. The infiltrating cells were confirmed to be T cells (line 139). The Immunohistochemical method used for CD3 staining has been added to line 472-484.

9. Data on Rejection Rates: In line 148, the statement that "MNB transplantation into fetuses yielded a reduced rate of rejection" lacks supporting data. Please clarify.

> We have demonstrated histological evidence of reduced rejection by comparing tissues where rejection was so severe that no glomeruli could be identified versus tissues where glomeruli and tubules remained intact. We have added "where glomerular and tubular structures could not be identified at all" on line 140 and modified the expression from "MNB transplantation into fetuses yielded a reduced rate of rejection" to indicate that "MNB transplantation into fetuses demonstrated attenuated rejection" on line 141.

10. CD31-Positive Images: Clarify how many experiments were performed and how many slides were stained in each experiment to ensure reproducibility and reliability of the results. Provide additional representative images if possible.

> We conducted experiments (n=2) to analyze chimeric vessels, randomly selecting two sections from each experimental tissue and evaluating the presence of chimeric vessels in 10 glomeruli per section. We have added these results to lines 148-151: "To evaluate the chimeric vessels, we assessed the proportions of recipient- and donor-derived vessels. GFP-positive/CD31-positive, i.e., donor-derived, vascular endothelial cells were observed in all glomeruli, whereas GFP-negative/CD31-positive, i.e., recipient-derived, vascular endothelial cells were detected in 45% of the glomeruli (Table 2)."

Additionally, we have added to lines 248-250: "GFP-negative/CD31-positive recipient-derived vascular endothelial cells were observed in all glomeruli, whereas GFP-positive/CD31-positive donor-derived vascular endothelial cells were detected in 30% of the glomeruli (Table 2 and Supplementary Fig. 2)."

We have also included Table 2 and Supplementary Figure 2 to support these findings.

11. Correct typo in Fig.2 legend: 2j should be replaced with 2k

> This has been corrected (line 651).

12. Line 197: Verification of the immunological advantages of fetal transplantation. This section describes a novel approach - two kidneys from the same fetal donor were transplanted into the same recipient at two different time points – fetal and adult. This should be better clarified in the title and in the text of this section. Please also better clarify the goal of this experiment.

> We have changed the title to "Attenuation of rejection reaction in secondary fetal kidney transplantation" (line 196).

We have added the following text to clarify the purpose (line 198-201): "To investigate whether immunological tolerance could be induced as one of the underlying mechanisms, the following experiment was conducted. One fetal kidney was transplanted during the fetal stage, whereas another fetal kidney from the same donor was cryopreserved and subsequently transplanted into the same recipient during the adult stage."

13. Clarify Data Presentation in Figure 4b-f: indicate which image corresponds to each experimental group, clearly describe the experimental groups, including retrieval times (two-week was compared to three week? – please clarify)

> We have added "11w retrieval (2weeks post-transplantation)" to Figure 4. Additionally, we have added the following text to the Results section (line 212-219): "In contrast, when GFP-SD rat MNBs were transplanted into adult SD rats (at 8 weeks of age) and tissue was retrieved 2 weeks after transplantation (Fig. 2h), glomerular and tubular structures could not be identified due to severe rejection. Therefore, the rejection reaction was attenuated in recipients who received a second fetal kidney transplantation after initial transplantation during the fetal stage (Fig. 4b–f), compared to those that received their first transplantation during the adult stage (Fig. 2h). However, the rejection reaction in the secondary transplantation, albeit attenuated, implied that complete immunological tolerance was not induced."

14. Separate and Describe Adult Transplantation Experiment (Lines 630-632): The description of the experiment involving transplantation into adult recipients should be clearly separated and presented under both the Methods and Results sections. This will ensure clarity and proper distinction from the fetal transplantation experiments.

> We have clarified the text within Figures 2d, f, and h by explicitly indicating "fetal-stage transplantation" and "adult-stage transplantation." Additionally, the method for transplantation into adult recipients has been described in the Methods section under "In the secondary fetal kidney transplantation performed during the adult stage" (line 432).

15. Fig 3d: please clarify that the line represents creatinine clearance and the bars represent urine volume. Provide details on how creatinine clearance was calculated. Include data on creatinine and urine volume measurements in the Methods and Results sections.

> We have specified in Figure 3d that the lines represent creatinine clearance and the bars represent urine volume. Additionally, we have added the following text (line 449-452) to describe the creatinine clearance calculation method: "Creatinine clearance was calculated using the following formula: Creatinine clearance [$\mu\text{L}/\text{min}$] = (urine creatinine levels [mg/dL] \times urine volume [$\mu\text{L}/\text{min}$]/serum creatinine levels [mg/dL])."

16. Fig 3h The comparison of day 150 with day 78 without aspiration punctures in Figure 3h represents a standalone experiment. Provide a detailed description of this experiment in both the Methods and Results sections, including the experimental design, sample sizes, and rationale for the comparison.

> We have added the following text to line 172-180 of the Results section: "To evaluate the efficacy of aspiration puncture in preventing hydronephrosis, we compared tissues with and without aspiration puncture. For the group without aspiration puncture, GFP-SD rat MNBs were transplanted into SD rat fetuses and retrieved 78 days posttransplantation ($n = 1$). The earlier timepoint of 78 days was selected as the approximate middle timepoint of the 150-day observation period, aiming to avoid severe hydronephrosis that would cause thinning of the renal parenchyma and lead to unclear histologic findings. Masson's trichrome staining revealed extensive fibrotic areas in the nonaspirated tissue (Fig. 3h). In contrast, the tissue retrieved with regular aspiration 150 days posttransplantation did not exhibit fibrotic areas ($n = 1$; Fig. 3h). These findings demonstrated that repeated aspiration punctures performed 1–2 times/week were sufficient to prevent hydronephrosis."

17. Fig 3i: Ensure consistent terminology throughout the text and figure legend (e.g., "storage time" versus "accumulation between punctures time"). Data for periods shorter than 24 hours do not appear on the plot. Please provide a more comprehensive explanation of the data in Figure 3i, both in the text and figure legend, to ensure readers fully understand the findings.

> We have standardized the terminology to "storage time" throughout the manuscript. Additionally, we have added the following text to the Figure 3 legend (line 669-670): "Because the minimum urine storage time between punctures was 24 h herein, no data points were available for <24 h ($n = 1$)."

18. Discussion: Discuss earlier reports, describing the ability of avascular, embryonic kidneys (1,2) and pancreas (3) to attract host vasculature, rendering them less susceptible to humoral

rejection (4):

- (1) Dekel, B. et al. Human and porcine early kidney precursors as a new source for transplantation. Nat. Med. <https://doi.org/10.1038/nm812> (2003).
- (2) Takeda, S. I., Rogers, S. A. & Hammerman, M. R. Differential origin for endothelial and mesangial cells after transplantation of pig fetal renal primordia into rats. Transpl. Immunol. <https://doi.org/10.1016/j.trim.2005.10.003> (2006).
- (3) Hecht, G. et al. Embryonic pig pancreatic tissue for the treatment of diabetes in a nonhuman primate model. Proc. Natl. Acad. Sci. U.S.A. <https://doi.org/10.1073/pnas.0812253106> (2009).

> We have incorporated the literature you suggested in the Discussion section. Reference (1) has been added as reference 33 on line 309-310. References (2) and (3) have been added as reference 40 and 41 on line 319.

2/6/2025

Manuscript number: COMMSBIO-24-4847A

Manuscript title: Fetal-to-Fetal Kidney Transplantation in Utero

Responses to Reviewers' Comments

We sincerely appreciate the reviewers' comments and suggestions, as well as their kind words. We have revised the manuscript to address these points. We believe that they have been improved through these revisions. Below are our responses to the comments. Responses to the comments are shown in red in the revised manuscript and in this letter.

Reviewer #1:

Thanks you for the revised version. The authors have addressed some of my concerns in the discussion. They have not added experimental data analyzing age of fetal transplant, host-donor vascularization and transplant rejection status but rather discuss this matter. They failed to cite important "old" papers that are highly relevant to the translation of fetal kidney transplantation clarifying how fetal kidney transplants sustain a molecular program that supports nephrogenesis in vivo and why fetal kidney transplants avoid immune rejection including Transplantation. 1997 Dec 15;64(11):1550-8, Transplantation. 2000 Apr 15;69(7):1470-8, J Am Soc Nephrol. 2002 Apr;13(4):977-990. These should be added.

> Thank you for your valuable comments and suggestions. We have incorporated the literature you suggested in the Discussion section. These three reference have been added as reference 34, 35, and 36 (line 309).

Reviewer #2:

The authors have extensively revised their manuscript within a quick turnaround time. The issues raised in the critiques from the three previous reviewers have been well addressed; including retitling the manuscript as suggested Reviewer 3. The Methods and Results sections have been better aligned with a much clearer separation of methods from the result sections. Greater technical detail has also been provided with: 1. clarification of the (n) within the experimental groups, 2. the software used for quantification of fluorescence, 3. more specific details of the experimental protocols used for the studies, 4. use of consistent terminology throughout the text, and 5. the inclusion of additional relevant literature. The edited manuscript reads more clearly along with improved graphs, tables and histological pictures. This is a well written manuscript on a topic that could hold potential for future clinical advancements.

> Thank you for your valuable comments. We will continue to dedicate ourselves to our research.

Reviewer #3:

Dear Editors,

Thank you for the opportunity to review this manuscript. The authors have thoroughly addressed my concerns and have made the necessary revisions to improve clarity and strengthen their work.

I believe the manuscript now meets the standards for publication and recommend acceptance.

> Thank you for your kind words. We will continue to dedicate ourselves to our research.